# Nanoscale imaging and control of altermagnetism in MnTe

O. J. Amin[1,12 ✉], A. Dal Din[1,12 ✉], E. Golias[2], Y. Niu[2], A. Zakharov[2], S. C. Fromage[1], C. J. B. Fields[1,3], S. L. Heywood[1], R. B. Cousins[4], F. Maccherozzi[3], J. Krempaský[5], J. H. Dil[5,6], D. Kriegner[7], B. Kiraly[1], R. P. Campion[1], A. W. Rushforth[1], K. W. Edmonds[1], S. S. Dhesi[3], L. Šmejkal[7,8,9,10], T. Jungwirth[1,11] & P. Wadley[1 ✉]

Nanoscale detection and control of the magnetic order underpins a spectrum of condensed-matter research and device functionalities involving magnetism. The key principle involved is the breaking of time-reversal symmetry, which in ferromagnets is generated by an internal magnetization. However, the presence of a net magnetization limits device scalability and compatibility with phases, such as superconductors and topological insulators. Recently, altermagnetism has been proposed as a solution to these restrictions, as it shares the enabling time-reversal-symmetry-breaking characteristic of ferromagnetism, combined with the antiferromagnetic-like vanishing net magnetization[1–4]. So far, altermagnetic ordering has been inferred from spatially averaged probes[4–19]. Here we demonstrate nanoscale imaging of altermagnetic states from 100-nanometre-scale vortices and domain walls to 10-micrometre-scale single-domain states in manganese telluride (MnTe)[2,7,9,14–16,18,20,21]. We combine the time-reversal-symmetry-breaking sensitivity of X-ray magnetic circular dichroism[12] with magnetic linear dichroism and photoemission electron microscopy to achieve maps of the local altermagnetic ordering vector. A variety of spin configurations are imposed using microstructure patterning and thermal cycling in magnetic fields. The demonstrated detection and controlled formation of altermagnetic spin configurations paves the way for future experimental studies across the theoretically predicted research landscape of altermagnetism, including unconventional spin-polarization phenomena, the interplay of altermagnetism with superconducting and topological phases, and highly scalable digital and neuromorphic spintronic devices[3,14,22–24].

For condensed-matter physics, the d-wave (or higher even-parity wave) spin-polarization order in altermagnets represents the sought-after, but for many decades elusive, counterpart in magnetism of the unconventional d-wave order parameter in high-temperature superconductivity[3]. For spintronics, altermagnets can merge favourable characteristics of conventional ferromagnets and antiferromagnets, considered for a century as mutually exclusive[3]. They can combine strong spin-current effects, which underpin reading and writing functionalities in commercial ferromagnetic memory bits, with vanishing net magnetization, enabling demonstrations of high spatial, temporal and energy scalability in experimental antiferromagnetic bits insensitive to external magnetic-field perturbations. These examples, as well as the predicted abundance of altermagnetic materials, ranging from insulators and semiconductors to metals and superconductors, illustrate the expected broad impact of this field on modern science and technology[3].

So far, however, the unconventional properties of altermagnets have been experimentally detected using spatially averaging electronic transport[4–11] or spectroscopy probes[12–19]. Here we report mapping of the altermagnetic order vector and demonstrate the controlled formation, from nanoscale to microscale, of a rich landscape of altermagnetic textures, including vortices, domain walls and domains. We use polarized X-ray photoemission electron microscopy (PEEM), which is a powerful tool in magnetism, allowing for, in addition to element specificity and magnetic sensitivity, concurrent full-field real-space imaging at the microscale with nanoscale resolution.

The measurements were performed at 100 K on a 30-nm-thick film of α-MnTe(0001) deposited on an InP(111)A substrate. Manganese telluride (MnTe) is one of the prototypical materials in altermagnetic research[2,7,9,12,14–16,18,20]. Below the transition temperature of 310 K, the magnetic order is within the $a$–$b$ plane of the film. The unit cell, shown

[1]School of Physics and Astronomy, University of Nottingham, Nottingham, UK. [2]MAX IV Laboratory, Lund, Sweden. [3]Diamond Light Source, Harwell Science and Innovation Campus, Didcot, UK. [4]Nanoscale and Microscale Research Centre, University of Nottingham, Nottingham, UK. [5]Photon Science Division, Paul Scherrer Institut, Villigen, Switzerland. [6]Institut de Physique, École Polytechnique Fédérale de Lausanne, Lausanne, Switzerland. [7]Institute of Physics, Czech Academy of Sciences, Prague, Czech Republic. [8]Max Planck Institute for the Physics of Complex Systems, Dresden, Germany. [9]Max Planck Institute for Chemical Physics of Solids, Dresden, Germany. [10]Institute of Physics, Johannes Gutenberg University, Mainz, Germany. [11]Present address: Institute of Physics, Czech Academy of Sciences, Prague, Czech Republic. [12]These authors contributed equally: O. J. Amin, A. Dal Din. ✉e-mail: oliver.amin@nottingham.ac.uk; alfred.daldin@nottingham.ac.uk; peter.wadley@nottingham.ac.uk

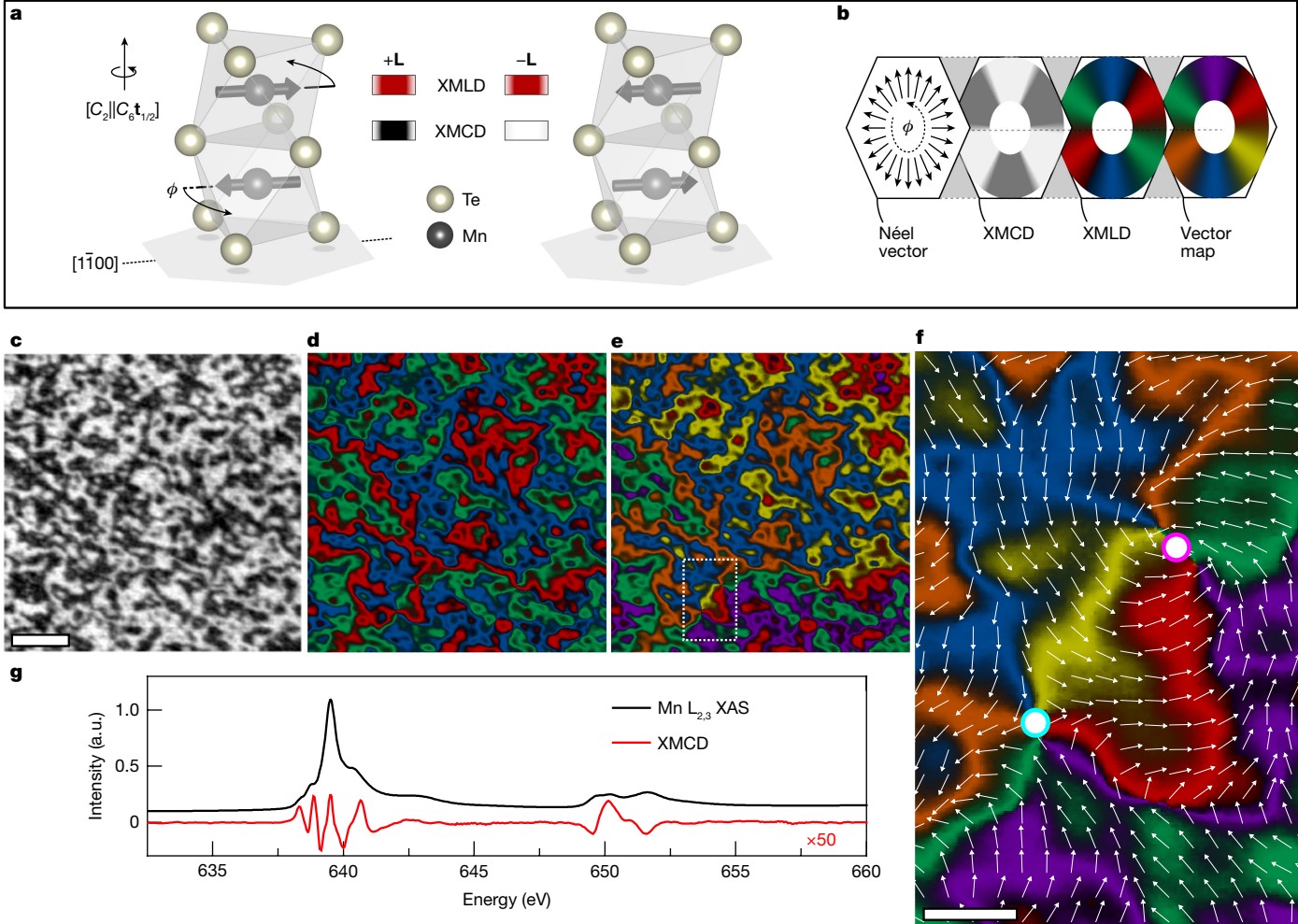

**Fig. 1 | Mapping of the altermagnetic order vector in MnTe. a**, Unit cell of α-MnTe with Mn spins collinear to the [1̄100] magnetic easy axis. Applying $\mathcal{T}$ transforms the left unit cell into the right. The unit cells with opposite **L** vector produce the same XMLD but inequivalent XMCD owing to $\mathcal{T}$-symmetry breaking in altermagnetic MnTe. **b**, Illustration of the vector mapping process. The colour wheels show the angular dependence of the XMCD, three-colour XMLD and six-colour vector map on the in-plane **L**-vector direction. The XMCD acts on the three-colour XMLD, with light XMCD regions changing the colour and dark XMCD regions leaving it unchanged to produce the six-colour **L**-vector map. In the XMLD and vector map, coloured segments indicate the magnetic easy axes oriented along the ⟨1̄100⟩ crystallographic directions.

**c**–**e**, XMCD-PEEM (**c**), XMLD-PEEM (**d**) and vector map (**e**) of a 25-μm² region of unpatterned MnTe film. **f**, An expanded view of the boxed region in **e** in which a vortex–antivortex pair is identified. The vortex–antivortex core positions are highlighted by the magenta–white and cyan–white circles, respectively. The combination of XMLD-PEEM and XMCD-PEEM imaging allows for unambiguous determination of the helicity of the swirling textures of the altermagnetic order vector, indicated by the six colours and overlaid vector plot. Scale bars, 1 μm (**c**) and 250 nm (**f**). **g**, X-ray absorption spectrum (XAS), plotted in black, and XMCD spectrum, plotted in red, measured across the Mn $L_{2,3}$ resonant edges. The XMCD spectrum is scaled by ×50. a.u., arbitrary units.

in Fig. 1a, contains two Mn atoms carrying magnetic moments $\mathbf{M}_1$ and $\mathbf{M}_2$ of equal magnitude and opposite direction. The two MnTe sublattices containing the opposite magnetic moments are connected by a spin symmetry combining a spin-space two-fold rotation with a real-space non-symmorphic six-fold screw-axis rotation ($[C_2\|C_6\mathbf{t}_{1/2}]$), and not by translation or inversion[2,7]. This non-relativistic spin symmetry of the crystal structure generates an altermagnetic (g-wave) spin polarization, which breaks the time-reversal ($\mathcal{T}$)-symmetry of the electronic structure[2]. The perturbative relativistic spin–orbit coupling generates a weak magnetization along the [0001] axis which, in zero external magnetic field, reaches a scale of only $10^{-3}\,\mu_B$ per Mn atom, where $\mu_B$ is the Bohr magneton[2,9,12].

## Mapping the local altermagnetic order

Our vector mapping includes the local real-space detection of the orientation of the altermagnetic order vector, $\mathbf{L} = \mathbf{M}_1 - \mathbf{M}_2$, with respect to the MnTe crystal axes in the (0001)-plane by X-ray magnetic linear dichroism (XMLD)-PEEM, and of the sign of **L** for a given crystal orientation by including X-ray magnetic circular dichroism (XMCD)-PEEM. In antiferromagnets with opposite spin sublattices connected by translation or inversion, the $\mathcal{T}$-odd XMCD is excluded by symmetry. In such cases, only the **L** axis can be detected by the $\mathcal{T}$-even XMLD-PEEM, but the sign of **L** remains unresolved[25–30]. Contrary to this, the recent theoretical and experimental spectroscopic study of altermagnetic MnTe has demonstrated the presence of a sizable XMCD, reflecting the $\mathcal{T}$-symmetry breaking in the electronic structure by the altermagnetic g-wave spin polarization[12]. Furthermore, the XMCD spectral shape owing to **L** pointing in the (0001) plane is qualitatively distinct from the XMCD spectral shape owing to a net magnetization $\mathbf{M} = \mathbf{M}_1 + \mathbf{M}_2$ along the [0001] axis[12]. This was demonstrated in ref. 12 by comparing the measured XMCD spectral shapes at a zero magnetic field and at a 6-T field applied along the [0001] axis. In the former case, **M** is weak and the measured spectral shape agrees with the predicted spectral shape due to **L**. In the latter case, **M** is sizable and qualitatively modifies the spectral shape, again in agreement with theory. We performed

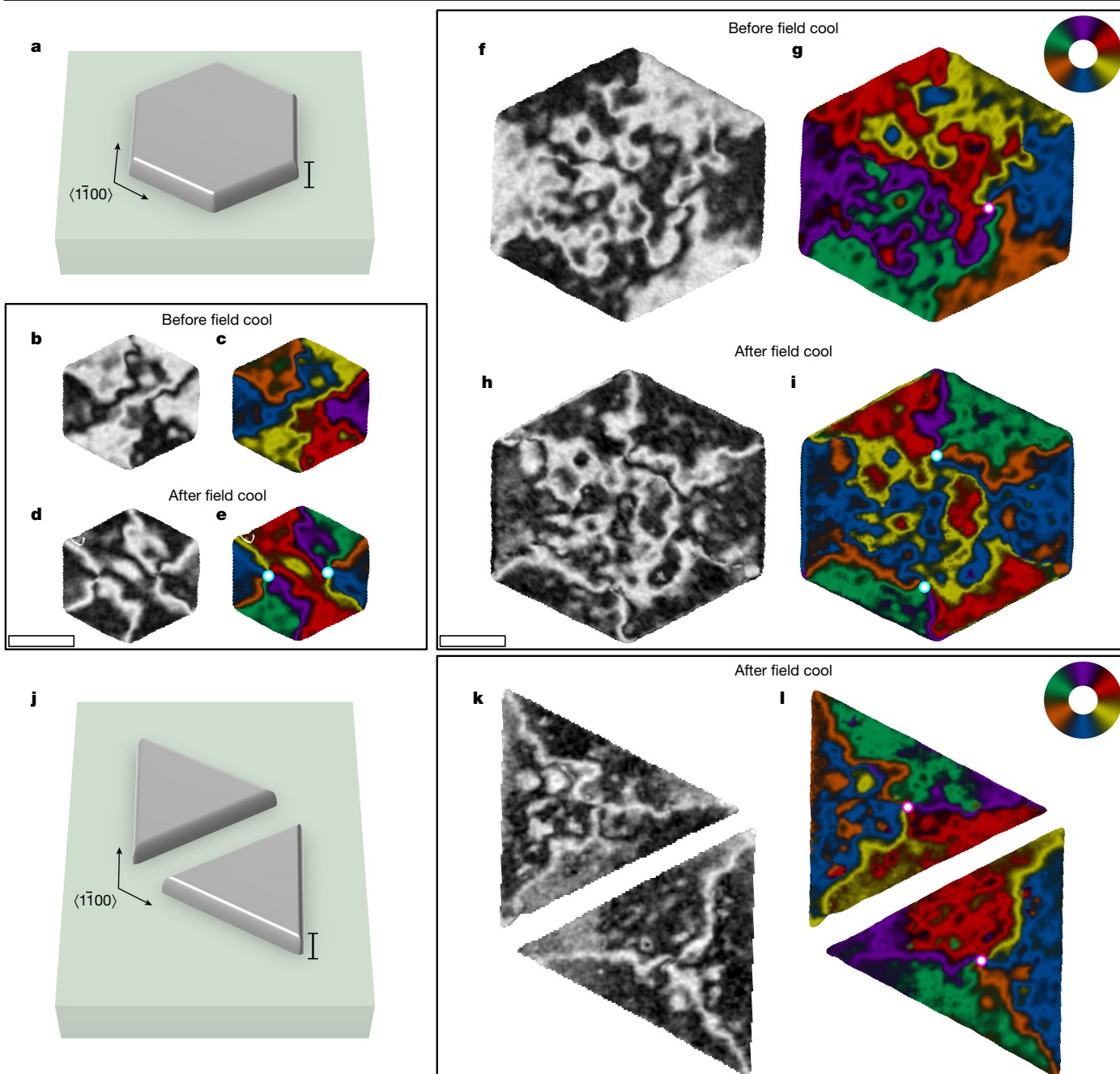

**Fig. 2 | Controlled formation of altermagnetic vortex nanotextures.**
**a**, Schematic of a hexagon microstructure with edges along the $\langle 1\bar{1}00\rangle$ axes. **b,c**, XMCD-PEEM map (**b**) and 6-colour vector map (**c**) of the virgin state of a 2-μm-wide hexagon. The **L**-vector axis preferentially aligns parallel to the hexagon edges with domain walls forming at the hexagon corners. **d,e**, The same as in **b** and **c**, respectively, but after cooling in a 0.4-T field applied along the [0001] axis, resulting in formation of only three domain types with 120° domain walls separating them at the hexagon corners. An antivortex pair forms in the centre of the structure, with core positions indicated by cyan–white circles. **f–i**, The same as in **b–e**, respectively, but for a 4-μm hexagon. **j**, Schematic of a pair of triangles with edges along the $\langle 1\bar{1}00\rangle$ axes. **k,l**, The same as in **d** and **e**, respectively, but for a pair of 4-μm triangle microstructures, with a single vortex at the centre of each structure indicated by the magenta–white circles. Scale bars, 30 nm (**a** and **j**),1 μm (**b–i**, **k** and **l**).

normal incidence X-ray PEEM, which is the optimum geometry for measuring both the in-plane Néel axis in the XMLD, and the altermagnetic XMCD. Images are taken at zero external field, where the XMCD signal owing to the weak relativistic remnant **M** is negligible compared with the altermagnetic XMCD owing to $\mathbf{L}\|\langle 1\bar{1}00\rangle$ directions in the (0001) plane[12]. The latter gives rise to our measured XMCD-PEEM contrast as confirmed by its spectral dependence (Methods and Extended Data Fig. 1). In analogy to the d.c. anomalous Hall effect, the XMCD can be described by the Hall vector, $\mathbf{h} = (\sigma_{zy}^{a}, \sigma_{xz}^{a}, \sigma_{yx}^{a})$, where $\sigma_{ij} = -\sigma_{ji}$ are

the antisymmetric components of the frequency-dependent conductivity tensor. For **L** in the (0001) plane of MnTe, **h** points along the [0001] axis, that is, $\sigma_{zy}^{a} = \sigma_{xz}^{a} = 0$ and $\sigma_{yx}^{a} \neq 0$, with the exception of $\mathbf{L}\|\langle 2\bar{1}\bar{1}0\rangle$ axes where $\sigma_{yx}^{a} = 0$ by symmetry.

The method of combining the XMCD-PEEM and XMLD-PEEM images into the vector map of **L** is illustrated in Fig. 1b. As the **L** vector subtends the angle, $\phi$, in the MnTe (0001) plane relative to the [$1\bar{1}00$] axis, the XMCD is proportional to $\cos(3\phi)$, with maximum magnitude for $\mathbf{L}\|\langle 1\bar{1}00\rangle$-axes and vanishing for $\mathbf{L}\|\langle 2\bar{1}\bar{1}0\rangle$ axes[12]. An XMCD-PEEM image

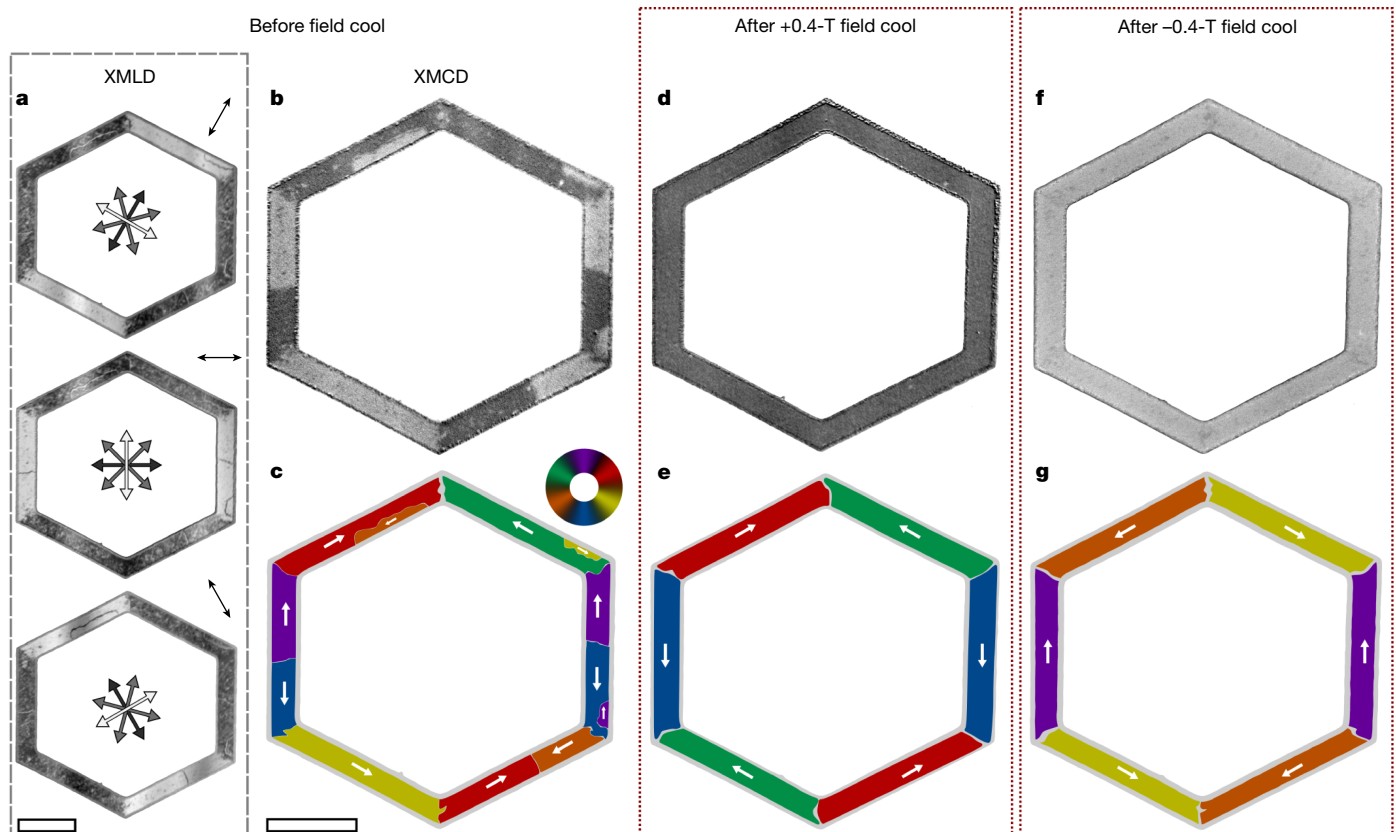

**Fig. 3 | Large single-domain altermagnetic states controlled by micropatterning and field cooling. a–g**, Images of an unfilled hexagon shape with arms, of 10 μm length and 2 μm width, aligned along the ⟨1$\bar{1}$00⟩ easy axes, before field cooling (**a–c**) and after field cooling with +0.4 T and −0.4 T (**d–g**). **a**, XMLD-PEEM images of the hexagon before field cooling for three directions of the X-ray linear polarization, indicated by the double-headed arrow in the top right corner of each image. The XMLD-PEEM contrast (double-headed arrows at the centre of each image) appears as light when **L** is perpendicular to the X-ray polarization, indicating large single spin axis domains in each arm, parallel to the arm edge. The 180° domain walls can be seen as thin, contrasting lines, separating domains with opposite direction of **L. b**, The corresponding XMCD-PEEM image reveals the direction of **L** along the spin axis parallel to the hexagon arms. **c**, A combination of the XMLD-PEEM and XMCD-PEEM images produces a six-colour vector map. The white arrows show the direction of **L** in the coloured domains. **d,e**, Repeat of **b** (**d**) and **c** (**e**) after field cooling the hexagon in a +0.4-T external magnetic field. **f,g**, Repeat of **d** (**f**) and **e** (**g**) after field cooling with the opposite-sign magnetic field. Scale bars, 5 μm.

of a 25-μm$^2$ unpatterned area of MnTe is shown in Fig. 1c, where positive and negative XMCD appear as light and dark contrast, respectively. The corresponding three-colour XMLD-PEEM map, shown in Fig. 1d, was obtained from a set of PEEM images taken with the X-ray linear polarization rotated, within the MnTe (0001) plane, in 10° steps from −90° to +90° relative to the horizontal [1$\bar{1}$00] axis. In this image, the local **L**-vector axis is distinguished (by red–green–blue colours), but the absolute direction remains unresolved. This information is included by combining the XMCD-PEEM and XMLD-PEEM in a six-colour vector map, shown in Fig. 1e,f, where positive XMCD regions change the colour (red–green–blue to orange–yellow–purple) of the XMLD-PEEM map and negative XMCD regions leave it unchanged. The Mn L$_{2,3}$ X-ray absorption and altermagnetic XMCD spectra are shown in Fig. 1g. The XMCD-PEEM images are obtained at fixed energy corresponding to the peak in the altermagnetic XMCD at the L$_2$ edge. The XMCD contrast reverses between positive and negative peaks of the XMCD spectrum, as shown in Extended Data Fig. 1, and vanishes at elevated temperatures where the spontaneous anomalous Hall effect is absent, as shown in Extended Data Fig. 2.

The characteristic vector mapping of **L** in our unpatterned MnTe film, shown in Fig. 1e,f, shows a rich landscape of (meta)stable textures akin to earlier reports in compensated magnets[26–30]. There exist 60° and 120° domain walls separating domains with **L** aligned along the different easy axes, as well as vortex-like textures. Highlighted in Fig. 1f is an example of an altermagnetic vortex–antivortex pair, analogous to magnetic textures previously detected in antiferromagnets such as CuMnAs (ref. 30). However, only the XMLD-PEEM was available in the antiferromagnet[30], that is, only the spatially varying Néel-vector axis could be identified, similar to our XMLD-PEEM image in Fig. 1d. In our altermagnetic case, we can add the information from the measured XMCD-PEEM (Fig. 1c). This allows us to plot the vector map of **L**, as shown in Fig. 1e,f. We directly experimentally determine that the **L** vector makes a clockwise rotation by 360° around the first vortex nanotexture, indicated by the magenta–white circle, whereas the other nanotexture is an antivortex with an opposite winding of the **L** vector, indicated by the cyan–white circle.

## Controlled formation of vortices

In Fig. 2, we show the designed formation of vortices with predetermined winding and position. We utilize a known edge effect, arising from an elastic energy term owing to magnetostriction of the film and film–substrate clamping, which can result in alignment of the **L** vector with respect to a patterned edge of a compensated magnet[31–34]. The edge effect is large enough to overcome the intrinsic magnetocrystalline anisotropy over a distance up to about 1.7 μm from the edge (Extended Data Fig. 3), where the length scale is governed by the interplay of anisotropy, exchange and destressing energies[34]. We leverage this by patterning, using electron beam lithography and argon ion

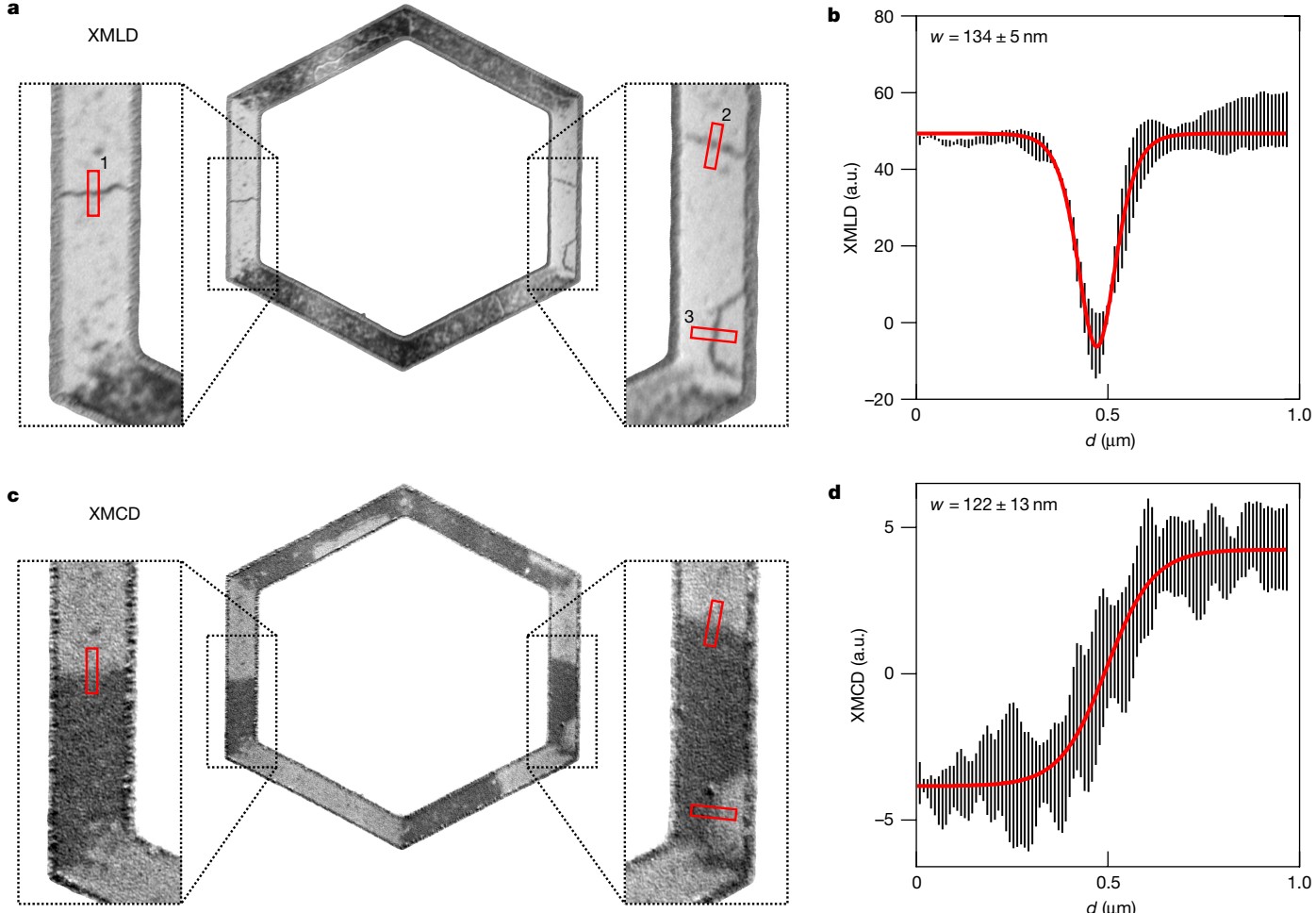

**Fig. 4 | The 180° domain-wall widths measured in the virgin state of an unfilled easy-axes hexagon with 2-μm-wide bars. a**, XMLD-PEEM image of the unfilled hexagon. Vertical bars containing 180° domain walls are shown as zoomed in insets. Line profiles across the domain walls are identified by red boxes labelled 1–3. **b**, Average domain-wall profile (black), measured in the XMLD and overlaid sech$^2$ fit line (red). The calculated domain-wall width is $w = 134 \pm 5$ nm. **c,d**, The same as in **a** and **b**, but measured in the corresponding XMCD-PEEM image. The average line profile, from dark to light domains, is fitted with a tanh function and the calculated width is $w = 122 \pm 13$ nm.

milling, MnTe structures of filled hexagon and triangle shapes with edges along the ⟨1Ī00⟩ easy axes.

In a virgin state, the interior of the hexagon splits into six wedge-shape domains with the **L**-vector axes aligned parallel to the hexagon edges, and with domain walls extending from the hexagon corners towards the centre of the structure (Fig. 2b,c). Two domains from opposite edges of the hexagon can have their **L** vectors parallel (one pair in Fig. 2b,c) or antiparallel (two pairs in Fig. 2b,c). In the next step, we select one sign of the **L** vector in each domain pair by first warming the structure above the MnTe magnetic transition temperature, and then cooling it back to 100 K in an external magnetic field of 0.4 T applied along the [0001] axis. In agreement with earlier spatially averaging measurements of the anomalous Hall effect and XMCD spectra[7,12], and explained by the coupling of the external field to **M** and of **M** to **L** (ref. 9), this procedure results in the population of only one sign of **L** in each pair of the ⟨1Ī00⟩ easy-axis domains (Fig. 2d,e). The formation of an antivortex pair in the centre of the hexagon is then required to resolve the total winding angle of the **L** vector through 720°. In Fig. 2f–i, we show analogous measurements in a larger hexagon. The observed magnetic configurations in the virgin state and after field cooling are similar to those in Fig. 2b–e near the hexagon edges, whereas in the central region they contain more complex textures reminiscent of the unpatterned film from Fig. 1.

In Fig. 2j–l, we show that the field-cooled state of triangle microstructures can stabilize isolated Bloch-type vortices, whose chirality is controlled by the triangle orientation. The different topological textures arise owing to the combination of the edge effect aligning the Néel vector parallel to the edge, and the external magnetic field selecting its sign. As the three edges of the triangle are 120°-separated, the **L** vector completes a total winding of 360°, which is facilitated by the formation of a single Bloch-type vortex. In Fig. 2k,l, mirrored triangle microstructures nucleate vortices with opposite chirality.

## Single-domain states

Moving from the nanoscale vortices to the opposite, large-scale limit of the real-space control and detection of the altermagnetic states, we show in Fig. 3 a designed formation of single-domain states in MnTe. Here we focus on a patterned unfilled hexagon shape with 10-μm arm length and 2-μm arm width and arms along the ⟨1Ī00⟩ easy axes. In the virgin state, the patterning alone generates large domain states with the axis of the **L** vector determined by the crystal direction of the hexagon arm. This is seen in the XMLD-PEEM images in Fig. 3a. The arms also show narrow 180° domain-wall lines with opposite contrast to the domains. In Fig. 3b, we show the XMCD-PEEM image of the hexagon and in Fig. 3c, we show the vector map obtained from the combined XMCD and XMLD-PEEM images. Regions within the hexagon arms where the XMCD-PEEM contrast reverses confirm the presence

of 180° domain walls separating opposite **L**-vector domains. Similarly, at the corners of the hexagon, XMCD-PEEM contrast reversal indicates 60° domain walls separating the **L**-vector domains in adjacent arms, and no contrast reversal indicates 120° domain walls.

To turn each arm into a micrometre-scale single-domain state, we apply the field-cooling procedure as in Fig. 2. The removal of the domain walls and the formation of the single-domain states in the arms is shown in the XMCD-PEEM image and vector map in Fig. 3d,e, respectively. In Fig. 3f,g, we show that reversing the direction of the magnetic field applied during cooling results in a reversal of the direction of **L** in each of the single-domain states. We show, in Extended Data Fig. 4, similar behaviour in a hexagon with 4-μm-wide arms, which represents the upper limit of device size to achieve single-domain states.

## Domain-wall profiles

In Fig. 4, we examine the domain-wall profiles in the zero-field-cooled state of the unfilled hexagon. For the XMLD and XMCD measurements, the dependence of the signal on distance $d$ across a 180° domain wall is described by functions $\mathrm{sech}^2(2d/w)$ and $\tanh(d/w)$, respectively. The domain-wall width parameter obtained for the fitted curves in Fig. 4b,d is $w = (134 \pm 5)$ nm for the XMLD image and $w = (122 \pm 13)$ nm for the XMCD image. Further analysis of domain-wall profiles in unpatterned regions is included in Extended Data Fig. 5.

## Outlook

The vector imaging and controlled formation of altermagnetic configurations ranging from nanoscale vortices and domain walls to microscale domains, demonstrated in this work, has broad science and technology implications. It is the basis on which the experimental field can develop, leveraging the $\mathcal{T}$-symmetry-breaking phenomenology, vanishing magnetization, ultrafast dynamics, and predicted compatibility of the altermagnetic order with the full range of conduction types from insulators to superconductors[3]. The X-ray dichroism vector mapping used here can be combined with other imaging techniques, such as X-ray laminography or ptychography, potentially offering depth sensitivity and even higher spatial resolution[35]. The ability to image and control the formation of microscale single-domain states will be highly relevant in the experimental research of fundamental electronic-structure properties of altermagnets, including the predicted unconventional non-relativistic and relativistic spin-polarization and topological phenomena, or interplay with other order parameters such as superconductivity[3,14,22–24]. Similarly, the controlled spatial uniformity of the altermagnetic states is an important step for the experimental research of digital spintronic devices. Multidomain states with spatially varying magnetic configurations represent a complementary area that can leverage the unique phenomenology of altermagnets in the research of topological skyrmions, merons and other magnetic textures, and in the related field of neuromorphic spintronic devices. Our demonstration of the vector mapping and controlled formation of the altermagnetic textures opens this experimental research front.

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

## Methods

### Sample fabrication

The 30-nm α-MnTe films used for this study were grown at about 700 K by molecular beam epitaxy (MBE) on InP(111)A substrates. The MnTe *c* axis was oriented parallel to the normal of the substrate surface. We confirmed the correct crystallographic phase and growth orientation of our MnTe films using X-ray diffraction, shown in Extended Data Fig. 6. Although MBE is a standard technique for growing epitaxial thin films, we note that sputtering has also been used to grow high-quality altermagnets, such as CrSb (ref. 17).

In this study, we present X-ray PEEM measurements on two epitaxial MnTe samples. Sample A (Fig. 1) was an uncapped α-MnTe film kept under ultrahigh-vacuum conditions and transported between the MBE and the PEEM in a custom-built vacuum suitcase. Sample B (Figs. 2 and 3) was an α-MnTe film capped with 2 nm of aluminium to prevent surface oxidation of the MnTe layer. We carried out microfabrication on sample B by coating with a 200-nm layer of polymethyl methacrylate (PMMA) photoresist then exposing by electron-beam lithography and developing in methyl isobutyl ketone (MIBK) mixed with isopropyl alcohol (IPA). Argon ion milling was used to fully remove the MnTe layer in the exposed areas before any residual resist was removed in acetone.

### PEEM imaging and Néel-vector mapping

The X-ray PEEM measurements were performed at the MAXPEEM beamline of the MAX IV Laboratory synchrotron. The X-ray beam was incident normal to the sample surface, with the X-ray linear polarization vector in-plane and the helicity vector out-of-plane. The linear dichroism asymmetry, XMLD $= (I(E_1) - I(E_2))/(I(E_1) + I(E_2))$, where $I$ is the measured pixel intensity, was calculated between images obtained at energies, $E_1$ and $E_2$, which correspond to maximum and minimum points in the magnetic linear dichroism spectra at the Mn $L_3$ absorption peak. The circular dichroism asymmetry, XMCD $= (I(\mu_+) - I(\mu_-))/(I(\mu_+) + I(\mu_-))$, was calculated between images obtained with opposite helicity polarizations, $\mu_\pm$, for a fixed energy corresponding to a maximum in the magnetic circular dichroism at the Mn $L_2$ absorption peak. The X-ray absorption spectroscopy and XMCD spectra shown in Fig. 1g were obtained at beamline I06-1 of Diamond Light Source, from a different chip cut from the same wafer of MnTe material.

XMLD maps were produced from dichroism asymmetry images with X-ray linear polarization at angles, $\theta = -90°$ to $\theta = 90°$, relative to the horizontal axis, in steps of 10°. The angular dependence of the XMLD was fitted with a $\sin(2(\theta + \varphi))$ function, where the phase offset, $\varphi$, encodes information about the local Néel-vector axis. The symmetry along the axis is broken by the XMCD, which is used as a mask to produce the vector maps. More details of the vector mapping process are included in Extended Data Fig. 7.

### Field cooling

Field-cooling cycles were done within the PEEM chamber at the MAX-PEEM beamline of the MAX-IV Laboratory synchrotron. Extended Data Fig. 8 shows a photograph of the set-up during field cooling. The sample was retracted to maximum distance from the microscope objective. A sample flag plate with attached permanent magnets was brought into proximity (about 300 μm) with the sample surface. We used 1.2-T neodymium–iron–boron magnets (N40EH) with dimensions of 12 mm × 12 mm × 3 mm, stacked in two pairs. We measured the field strength, normal to the sample surface, at about 300 μm to be 0.45 T. The sign of the field was reversed by flipping the permanent magnet flag plate.

To carry out a field-cool cycle, we heated the sample using a filament on the sample holder to 350 K. This was above the 300-K Néel temperature of our samples. With the permanent magnet in proximity to the sample surface, we cooled the sample from 350 K to 100 K using liquid nitrogen.

### Analysis of easy- and hard-axes domains

XMLD- and XMCD-PEEM images of the easy-axes and hard-axes hexagons after zero field cooling are shown in Extended Data Fig. 9. The XMLD-PEEM images were taken with the X-ray linear polarization collinear to the horizontal axis of the image. Light and dark contrast corresponds to in-plane Néel domains aligned perpendicular and parallel to the X-ray linear polarization, respectively.

From the regions of single contrast in the XMLD-PEEM images, we determined that the device patterning aligns the Néel vector parallel to the edge, and that the 2-μm bar width is narrow enough to induce large single domains. A comparison between the hexagon patterned with edges parallel to the MnTe magnetic easy axes (Extended Data Fig. 9a) and hard axes (Extended Data Fig. 9c) reveals a similar domain morphology, from which we conclude that the magnetic anisotropy induced by the edges is dominant over the intrinsic magnetocrystalline anisotropy of the MnTe film.

The XMCD-PEEM image of the easy-axes hexagon (Extended Data Fig. 9b) shows clear dark and light domains, which are well correlated with the domain walls observed in the corresponding XMLD image (Extended Data Fig. 9a). For the hard-axes hexagon, the contrast in the XMCD image is significantly weaker with a much smaller length scale. This is as expected as the XMCD is disallowed by symmetry when the magnetic moments are aligned with the $\langle 2\bar{1}\bar{1}0 \rangle$ axes[12]. The distribution histogram of the XMCD-PEEM image pixel values within the outlined regions (blue area of Extended Data Fig. 9b and red area of Extended Data Fig. 9d) is shown in Extended Data Fig. 9e. The small XMCD contrast visible in Extended Data Fig. 9d most likely arises from small local variations in the magnetic moment orientation.

## Data availability

The data supporting the findings of this study are available from the corresponding authors upon request.

**Acknowledgements** We thank MAX IV Laboratory for time on Beamline MaxPEEM under proposal 20231714 (O.J.A.). Research conducted at MAX IV, a Swedish national user facility, is supported by the Swedish Research Council under contract 2018-07152, the Swedish Governmental Agency for Innovation Systems under contract 2018-04969, and Formas under contract 2019-02496. We thank Diamond Light Source for the provision of beamtime under proposal number MM36317. Electron-beam lithography was carried out at the nanoscale and microscale research centre supported by EPSRC Grant P/M000583/1. O.J.A. acknowledges support from the Leverhulme Trust Grant ECF-2023-755. D.K. acknowledges the Czech Science Foundation (Grant 22-22000M) as well as Lumina Quaeruntur fellowship LQ100102201 of the Czech Academy of Sciences. L.S. acknowledges funding by the Deutsche Forschungsgemeinschaft (DFG, German Research Foundation)-TRR288-422213477 (projects A09 and B05). T.J. acknowledges the Ministry of Education of the Czech Republic Grant CZ.02.01.01/00/22008/0004594 and ERC Advanced Grant 101095925. P.W. acknowledges support from the Royal Society through a University Research Fellowship. The work was supported by the EPSRC grant EP/V031201/1.

**Author contributions** O.J.A., A.D.D., K.W.E., S.S.D., L.S., T.J. and P.W. conceived and led the project. O.J.A., A.D.D., R.P.C., S.L.H., R.B.C. and A.W.R. contributed to growth and fabrication of materials and devices. O.J.A., A.D.D., P.W., K.W.E., B.K., C.J.B.F., S.C.F., E.G., Y.N., A.Z., S.S.D. and F.M. performed the XPEEM experiments and data analysis. O.J.A., A.D.D., P.W., K.W.E., C.J.B.F., S.C.F., S.S.D., D.K., J.K. and J.H.D. performed sample characterization. P.W., T.J., S.S.D., O.J.A., A.D.D. and K.W.E. wrote the paper with feedback from all authors.

**Competing interests** The authors declare no competing interests.

**Additional information**
**Correspondence and requests for materials** should be addressed to O. J. Amin, A. Dal Din or P. Wadley.

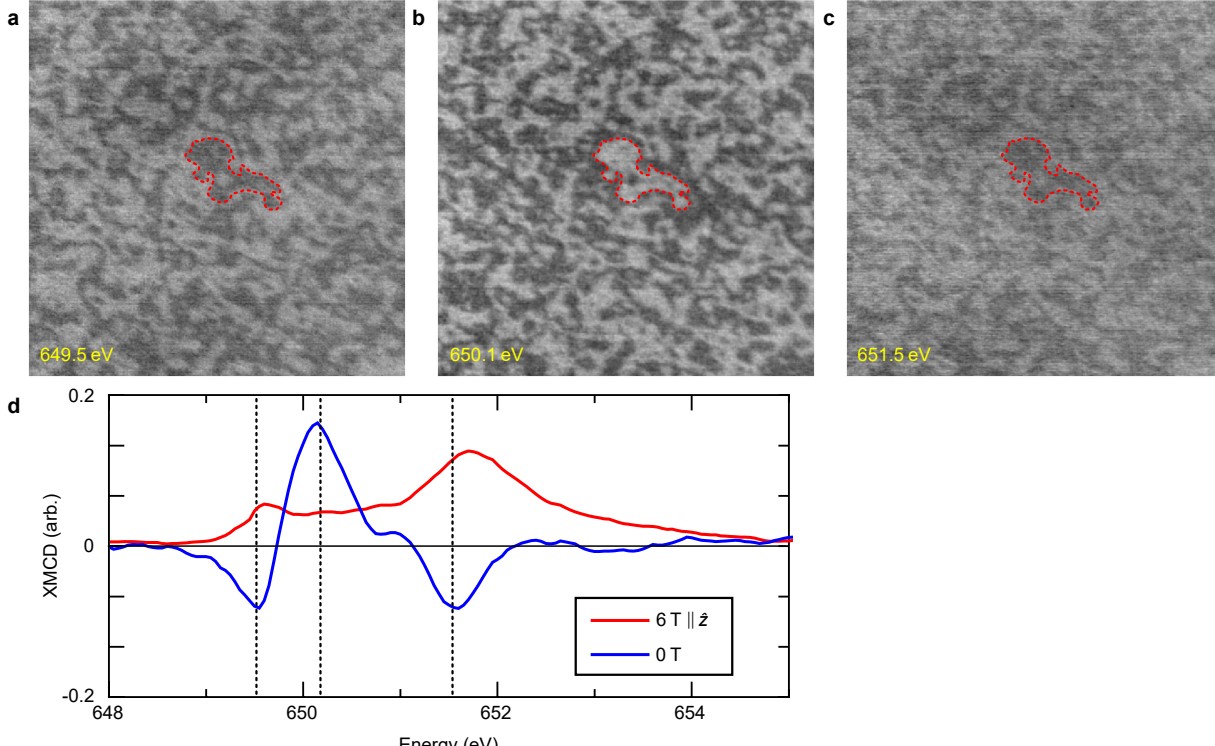

**Extended Data Fig. 1 | XMCD reversal across the Mn $L_2$ resonance edge. a-c**, XMCD-PEEM images measured at 649.5 eV, 650.1 eV, and 651.5 eV, respectively. The red outline highlights a domain in the centre of the image to aid the viewer in identifying the contrast reversal. **d**, Mn $L_2$ XMCD spectra in zero field (blue) and at 6 T (red), taken from ref. 12. The vertical dashed lines indicate the peaks in the zero field XMCD, where the XMCD-PEEM images in a-c were recorded. The zero field XMCD reverses sign between the three different energies, consistent with a-c, which the 6 T XMCD has positive sign across the whole $L_2$ multiplet.

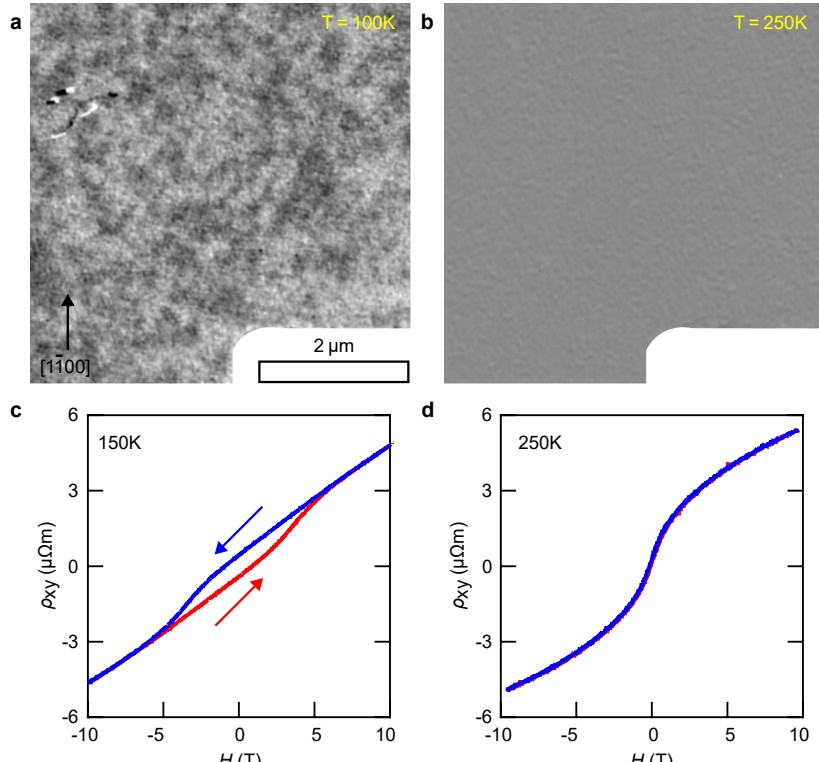

**Extended Data Fig. 2 | XMCD-PEEM images taken at low and high temperature. a**, XMCD-PEEM image taken at temperature T = 100 K, showing magnetic contrast in open-space region in proximity to a patterned edge. **b**, The same region re-imaged at T = 250 K, where the XMCD magnetic contrast vanishes, leaving only structural contrast. **c, d**, Hall resistivity measurements on a 100 μm Hall bar at T = 150 K and 250 K, respectively, showing that the spontaneous anomalous Hall effect is absent at the higher temperature.

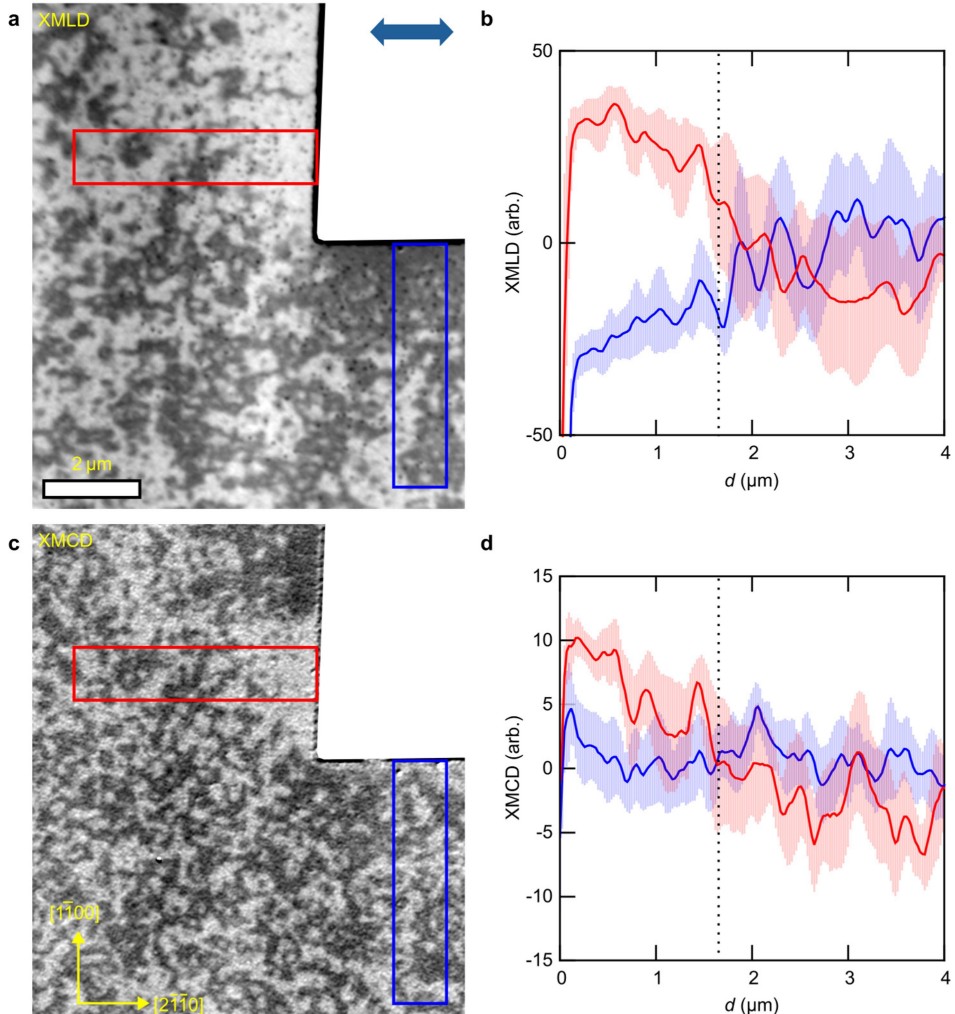

**Extended Data Fig. 3 | Length scale of domains induced by patterned edges along magnetic easy and hard directions. a**, XMLD-PEEM image, taken with X-ray polarization parallel to the horizontal axis, of an open space region of MnTe in proximity to a patterned corner edge. **b**, Line profile measurements of the XMLD as a function of distance, $d$, from the patterned edge, parallel to the ⟨1$\bar{1}$00⟩ magnetic easy axis (red) and ⟨2$\bar{1}\bar{1}$0⟩ magnetic hard axis (blue). Solid lines are the average line profiles measured within the boxed regions in (a), with the standard deviation plotted as an envelope. The dashed vertical line indicates the nucleation length (1.7 μm) of the edge-induced domain, where the XMLD becomes comparable for easy and hard axis edges. **c, d**, Same as a, b, respectively, for the corresponding XMCD-PEEM image.

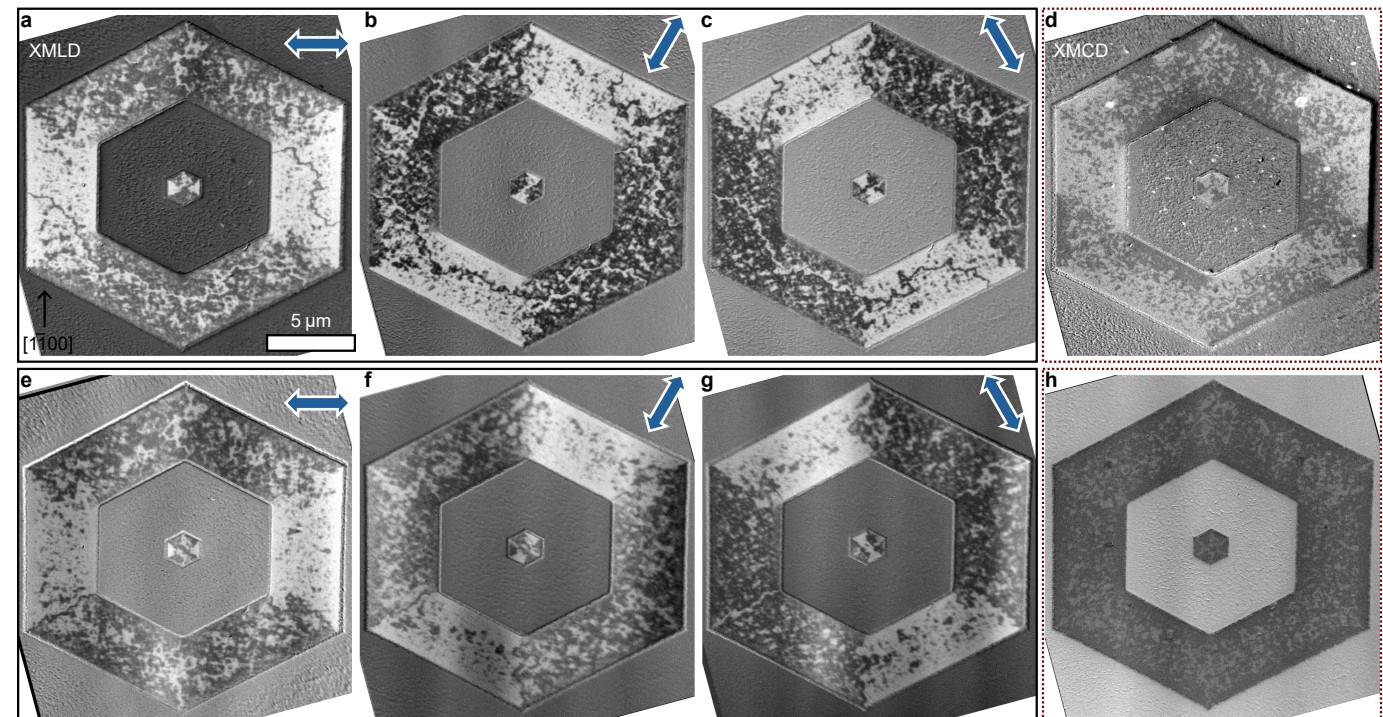

**Extended Data Fig. 4 | Unfilled hexagon with 4 μm wide bars patterned parallel to the ⟨1Ī00⟩ MnTe magnetic easy axes. a-c**, Virgin state XMLD-PEEM images taken with X-ray linear polarization (blue double-headed arrow) 0°, 60°, 120° to the horizontal axis, respectively. **d**, Virgin state XMCD-PEEM image. **e-g**, Same as a-c, for field-cooled state. **h**, Field-cooled state XMCD-PEEM image.

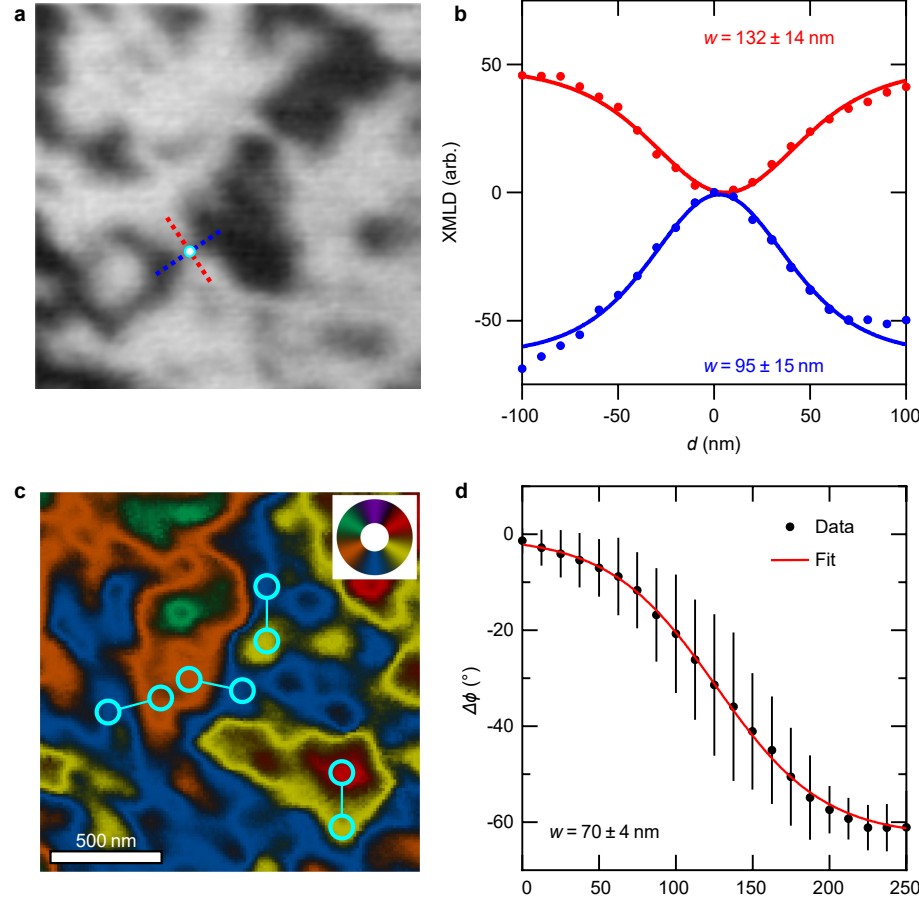

**Extended Data Fig. 5 | Determining vortex and 60° domain wall widths.**
**a**, XMLD-PEEM image of vortex-antivortex. The position of the cyan and white
circle is at the centre of the antivortex. Orthogonal line profiles are taken
following the red and blue dashed lines. **b**, XMLD measured along the red and
blue line profiles. A sech$^2$ fit gives the vortex profile width as $w = 132 \pm 14$ nm
(red) and $w = 95 \pm 15$ nm (blue). **c**, Colour map of in-plane Néel vector direction.
Highlighted by connected circles are line profile locations traversing 60°
domain walls between coloured $\langle 1\bar{1}00 \rangle$ easy-axes domains. **d**, Plot of average
phase variation, $\Delta\phi$, across 60° domain walls (black) and tanh fit line (red). The
measured domain wall width is $w = 70 \pm 4$ nm.

**a**

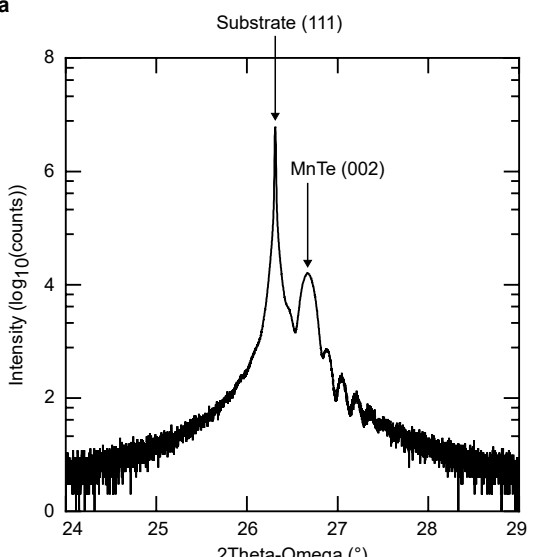

**b**

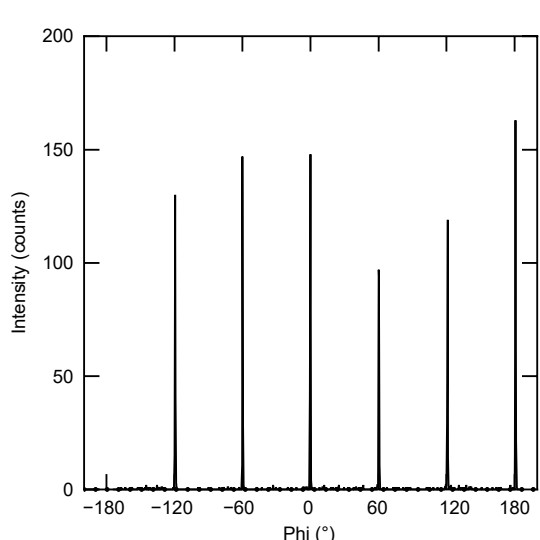

**Extended Data Fig. 6 | X-ray diffraction measurements of MnTe epitaxially grown on InP(111)A substrate. a**, 2Theta-Omega scan showing MnTe *c*-axis (002) peak relative to substrate (111) normal. **b**, Phi scan centred on MnTe (012), showing six-fold in-plane symmetry corresponding to $\alpha$-MnTe phase with NiAs structure.

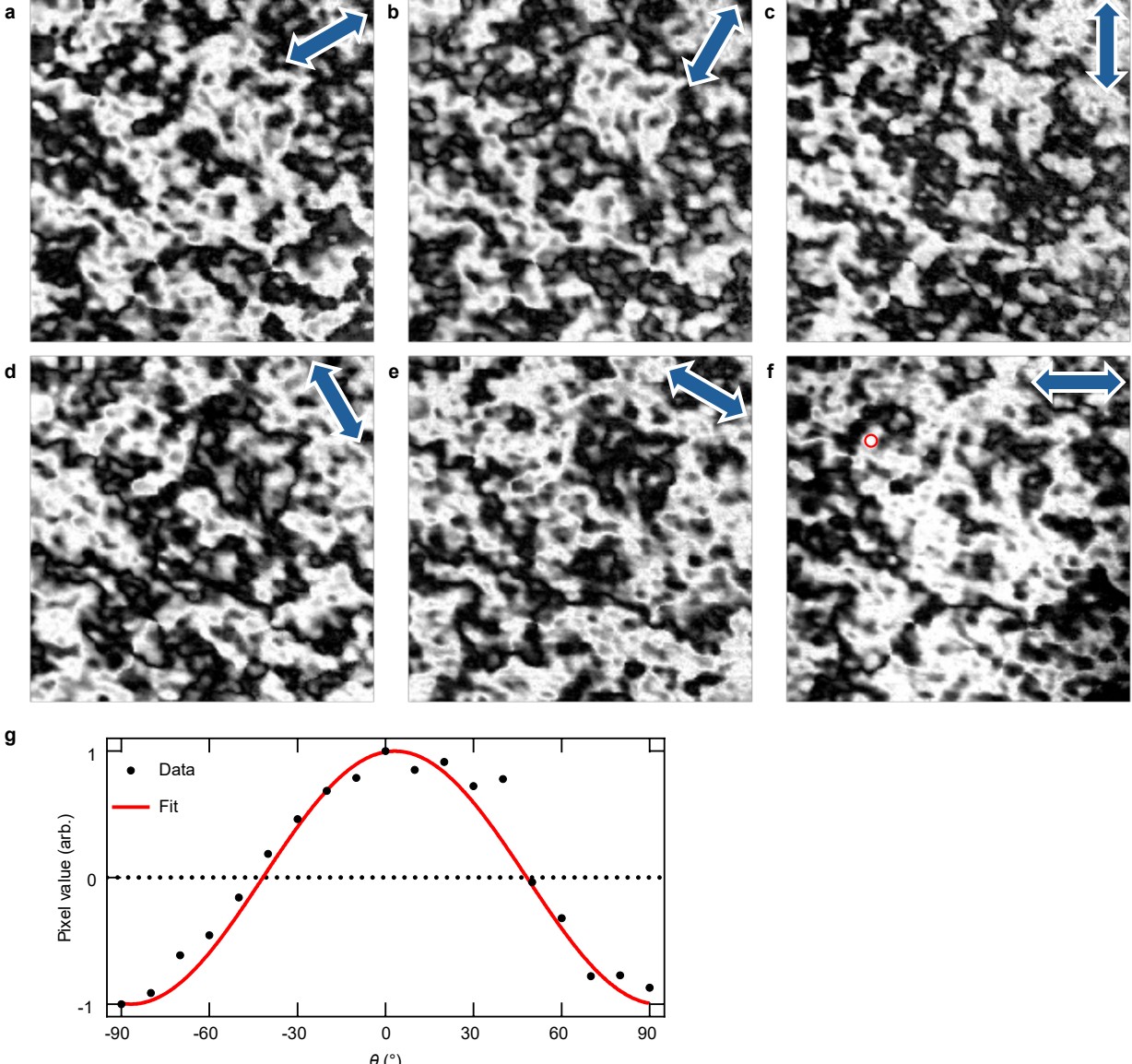

**Extended Data Fig. 7 | Calculating the Néel vector axis from XMLD-PEEM images with rotated X-ray linear polarization vector. a-f**, Normalised XMLD-PEEM images of the open-space region presented in Fig. 1 of the main text, for X-ray linear polarization vector (blue double-headed arrow) at −30°, −60°, 90°, 60°, 30°, and 0° to the horizontal axis. **g**, Normalised XMLD intensity (of pixel circled in f), for the full set of X-ray linear polarization angles, $\theta$, between −90° and 90° in steps of 10°. The $\sin(2(\theta + \phi))$ fit encodes information about the local Néel vector axis, $\phi$. In this example, $\phi = 41.8°$. Repeating the fitting process for every pixel location produces the XMLD map shown in Fig. 1d of the main text.

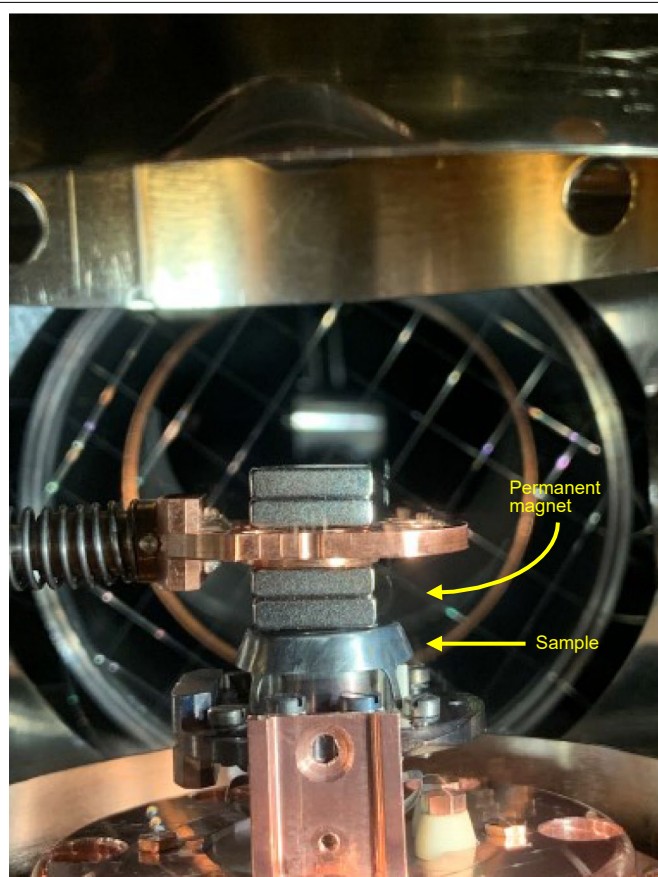

**Extended Data Fig. 8 | *In situ* field cooling set-up at MAXPEEM.** Photograph of the field cool set-up seen through a viewing port of the PEEM chamber. The permanent magnet is held in proximity with the sample during temperature cycling.

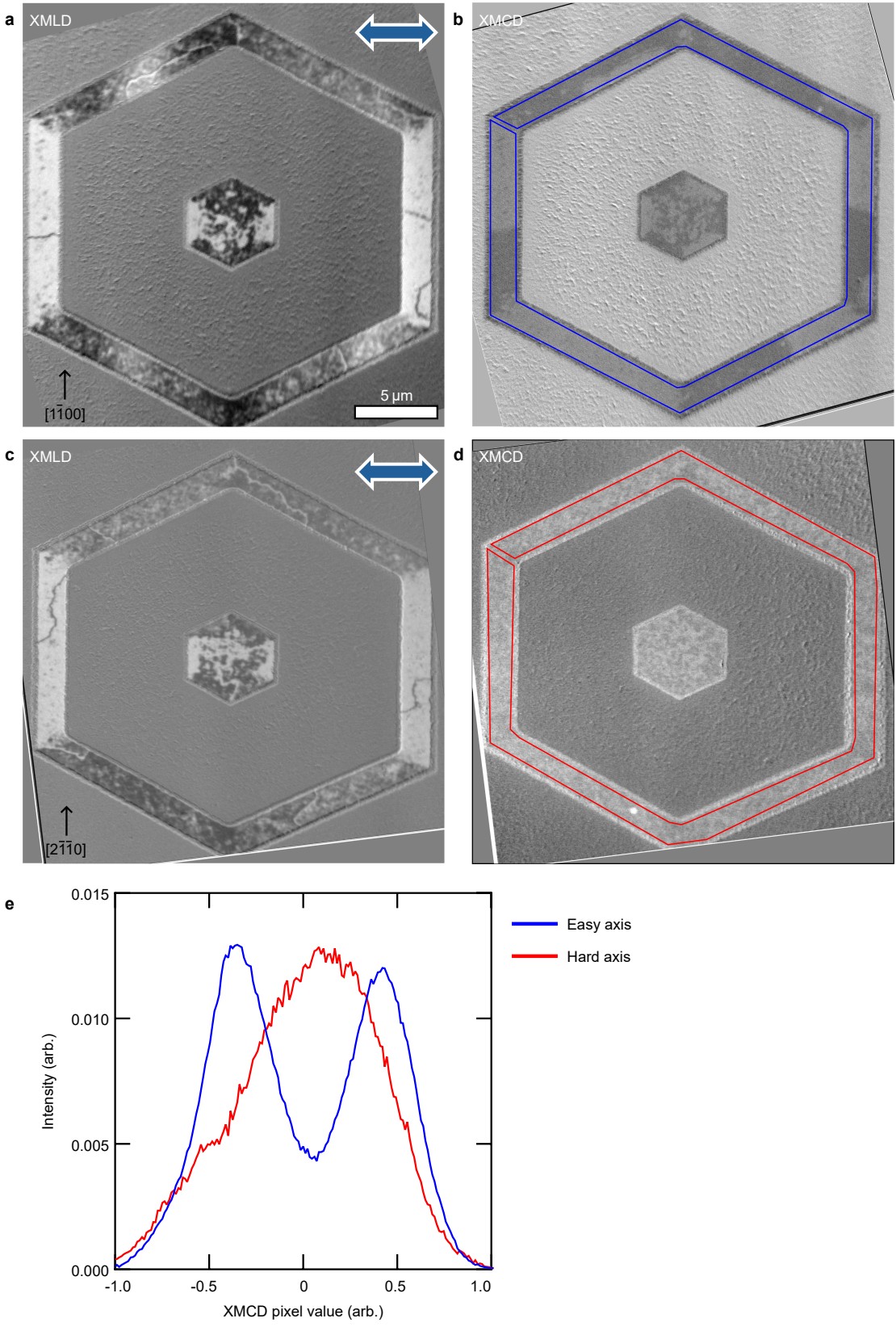

**Extended Data Fig. 9 | Pattern-induced domain formation in 2 μm bar width unfilled hexagons with edges aligned parallel to the ⟨1̄100⟩ magnetic easy axes and the ⟨21̄1̄0⟩ magnetic hard axes. a, b**, XMLD- and XMCD-PEEM images, respectively, of easy axis hexagon. **c, d**, Same as a, b, for hard axis hexagon. **e**, Intensity distribution of XMCD values measured in the blue and red outlined regions in b and d.