## [Peer Review File · Nature]

Nanoscale imaging and control of altermagnetism in MnTe

Corresponding Author: Dr Oliver Amin

Version 0:

Reviewer comments:

Referee #1

(Remarks to the Author)

This manuscript presents spatially-resolved, full-vector imaging of altermagnetic domains in alpha-MnTe by combining XMLD-PEEM with XMCD-PEEM. As well explained in the manuscript, the Néel order of collinear antiferromagnets does not exhibit XMCD contrast due to time reversal symmetry. In contrast, the order parameter of altermagnets, which breaks time reversal symmetry, allows for the detection and full-vector imaging of the direction of the order parameter. This key distinction between collinear antiferromagnets and altermagnets has enabled the authors to image various spin textures, such as vortices, anti-vortices, and their pairs. Additionally, the authors have demonstrated clear domain wall formation and achieved a single domain state through field cooling in unfilled hexagon samples.

An important feature of altermagnets is their antiparallel alignment of atomic moments in real space, akin to collinear antiferromagnets, combined with spin-split bands in momentum space, characteristic of ferromagnets. This unique combination offers both advantages of antiferromagnets (i.e., no net magnetic moment and THz dynamics) and ferromagnets (i.e., directional spin polarization), which was mutually exclusive before. As a result, the altermagnetism attracts significant interest across magnetism, spintronics, and broader fields of condensed matter physics. Previous studies have identified fundamental characteristics of altermagnets such as the anomalous Hall effect, spin splitter torque, and spin-split bands. These features critically depend on altermagnetic domain structures, as different domains produce different signals with varying signs, leading to signal cancellation in multi-domain structures. Thus, investigating altermagnetic domain structures and developing methods to achieve single-domain states are the most important next steps for altermagnet studies. This work convincingly demonstrates imaging of altermagnetic domain structures and, importantly, methods to achieve domain wall and single domain states, providing a significant advancement towards future research on altermagnetic dynamics and potential applications. Moreover, the manuscript is well-written and organized. In this respect, I support publication of this manuscript Nature after the authors address the following comments in revision:

1. Please provide more details on the influence of magnetoelastic forces and surface anisotropy in aligning the L-vector with respect to a patterned edge, as stated. Understanding this mechanism is crucial for explaining the difference in domain structures between smaller and larger hexagon samples (Fig. 2), as well as the formation of the single domain state (Fig. 3). Additionally, please clarify whether the orientation of altermagnetic domains is determined by the edge direction rather than the magnetic easy axis of alpha-MnTe. What implications arise if the edge direction differs from the magnetic easy axis?

2. I assume that the alpha-MnTe imaged in this study is a single crystal, both in-plane and along the thickness direction. This is important as the order parameter of an altermagnet is magnetic multipole, which results from the product of electric multipole with spin, forming a ferroic order that breaks time reversal symmetry and enables XMCD imaging. The orientation of the electric multipole directly correlates with the crystallographic orientation, limiting the size of altermagnetic domains below the size of crystallographic grains. For practical applications, using sputtering (unlike the MBE method employed in this work) is common but introduces many small grains with random orientations, complicating the realization of well-defined domain walls and single domain states. While this does not detract from the novelty and significance of the current work, a discussion on this issue would motivate the reader to investigate a way of achieving well-defined domain wall and single domain states in sputtered altermagnets, which is of crucial importance for applications.

Referee #2

(Remarks to the Author)

A. Summary of the key results

The MS by Amin et al reports an experimental study of a thin film of α -MnTe (0001) which is supposed to be an altermagnetic material. The sample was epitaxially grown on an InP(111) substrate and the signature features of an altermagnet was validated using XMCD and XMLD with an X-PEEM.

The topic of altermagnetism is very timely as this new class of magnetic materials should change the entire paradigm of how to look at magnetism. Altermagnetism (AM) combines features of antiferromagnetism (AF), i.e. no net magnetic moment with features from ferromagnetism (FM), i.e. spin ordering in q -space. Since FM and AF exhibit different T-reversal symmetry, this is scientifically a fascinating topic and might lead to unprecedented opportunities with future magnetic devices.

The approach by this study is based on a combination of the XMCD effect, which is sensitive to FM, i.e. magnetization $\langle M \rangle$, and with XMLD, which is known to be primarily sensitive to AF ordering, i.e. $\langle M^2 \rangle$. The microscopic magnetism of the MnTe film is addressed by using both dichroism effects as contrast mechanism in a spatially resolving state-of-the-art X-PEEM instrument at MAX-IV, where the alignment of the x-ray beam and the sample orientation can be varied. The authors also demonstrate the modulation of the AM state via external fields and temperature in a micropatterned system. It is important to note that the authors have published a preliminary magnetic X-ray spectroscopy study with this material (not clear whether it was the exact sample), where they observed a crystal orientation dependent XMCD signal, which is an indicator for the k -dependent spin polarization in an altermagnetic system.

B. Originality and significance: if not novel, please include reference

The work presented is highly novel and of great interest to a broad audience, therefore considering it for a publication in Nature is justified. The fact that the first milestone, i.e. the non-spatially resolved XMCD spectroscopy study was published previously in PRL, is not necessarily a weakness, although it could have been combined into a single high profile publication

C. Data & methodology: validity of approach, quality of data, quality of presentation

The data quality is high, but I have a concern regarding the validity of the approach. The depth resolution of X-PEEM is limited and therefore it is unclear whether the surface properties impact the conclusion about the altermagnetic phase. Also, most of the theory work on AM materials seems to be bulk focused and it would be interesting to know, whether the thickness of the sample and the above mentioned depth limited sensitivity of X-PEEM could have an impact to the conclusion. In other words, can we expect that any bulk crystal that shows AM behavior can equally be fabricated in thin film form and still maintain the AM character. For the technological applications of AM materials, this would be of significant importance.

Regarding the presentation itself, I have several comments that need to be addressed before I can recommend it for publication in any peer-reviewed journal

- The abstract is confusing and does not reflect the actual work. It needs to be rewritten
- There are several instances of "generic phrases", which I recommend to reconsider. Examples are "abundance of robust altermagnetism" (is this scientifically justified?), "offering the optimal combination of microscale field of view and nanoscale detection limit". I would question this statement entirely. There are several other examples in the text that I hope that authors will be able to either provide clarity or rephrase.
- P4: "full-vector mapping" It is important to point out that the 3D vector magnetization is only obtained (if at all) in the detectable depth of X-PEEM. And, even there, I doubt that the sensitivity is depth-independent, which means that other techniques, such as X-ray laminography, where the same contrast mechanisms can be employed, might provide a more accurate mapping.
- Figure captions/ labelling: The authors need to provide scale bars and statements about the spatial resolution in the obtained images. The color wheel in Fig 1b is too small, it should be enlarged, since this is the essential information to follow the conclusions.
- Fig 1c: Is this really all magnetic contrast? What is the sample roughness?
- Fig 1f: what are the black rims surrounding the colored areas? The two circles indicating the V-AV pair are confusing. I suggest to consider a closer zoom-in for those regions.
- Fig 2: what are the gray areas surrounding the magnetic domains?
- Fig 3: Micropatterning the AM system: how relevant are the length scales of the microstructure

D. Appropriate use of statistics and treatment of uncertainties

The analysis of the data is state-of-art, however, as mentioned above a critical assessment of the limitations of the X-PEEM technique as such and mentioning potential alternatives is missing

E. Conclusions: robustness, validity, reliability

The conclusions drawn are consistent within the limits of the experimental sensitivities of X-PEEM.

F. Suggested improvements: experiments, data for possible revision

See detailed comments under C:

G. References: appropriate credit to previous work?

The list of references is appropriate, and the authors are commended for including even all the arXiv papers in this rapidly evolving field

H. Clarity and context: lucidity of abstract/summary, appropriateness of abstract, introduction and conclusions

See detailed comments under C:

In summary, I find this work highly interesting, very timely and relevant to the current discussion in the magnetism community about AM as a new class of magnetic materials. Questions that need to be addressed are the general validity of AM in thin films, which would be most relevant to technological applications. The discussion about topological features, i.e. the V-AV pair could be expanded.

Before I can recommend it for publication, the authors need to address my comments above.

Referee #3

(Remarks to the Author)

Amin et al present the imaging of the magnetic configuration of the altermagnetic material MnTe. Through the combination of

the X-ray magnetic circular and linear dichroism, which probe time-reversal symmetry breaking, and the Neel orientation, respectively, they map the altermagnetic vector field in a 30 nm thick MnTe film, revealing the presence of vortices, antivortices, and domain wall textures. They demonstrate that by field cooling, they can alter the magnetic configuration. In addition, they go on to demonstrate that by patterning the film into micro-hexagons and microwires, they can achieve local ordering of the Neel vector.

Altermagnetism is currently a very “hot topic” in the condensed matter physics community, with many theory papers, and a growing number of experimental works being published in recent years. This work represents the first spatially resolved measurements of the magnetic configuration of an altermagnet, providing insight into the types of textures that form, as well as the influence that microstructuring can have on the magnetic configuration. The work is very timely, and I'm sure will be of interest to many in the community.

However, there are two main reasons why I believe that this manuscript is not suitable for publication in Nature:

1. I find the results are oversold.

The title “Altermagnetism imaged and controlled down to the nanoscale” does not reflect the achievements of the paper. Yes, the manuscript presents the imaging of altermagnetic domains at the nanoscale. However, the title is misleading in that it implies that the magnetic configuration can be controlled at the nanoscale, which is not the case. The authors show that they can create micro-metre-sized domain structures with micropatterning. However, it is not the case that the state can be precisely or locally *controlled* at the nanoscale, but rather that they observe the influence of surface effects and magnetoelastic effects on the local magnetic configuration.

In terms of the field of altermagnetism itself, there are many claims in the first paragraph of the main text, and the authors give a misleading impression that the field is mature and well established. However, only one perspective article is cited. For example, they claim there is an “abundance of robust altermagnetism among materials ranging”. I would disagree with this: the field is still very young, and while very exciting, is still developing. The authors should soften their claims in the introduction of the manuscript, and provide a more balanced discussion.

The authors mention multiple times that they achieve a “full-vector mapping”. However, they only map the vector in the plane. A full vector mapping would be three dimensional.

The authors mention many times “control of configurations”: e.g. “demonstrate the control, from nano- to microscale, of a rich landscape of altermagnetic textures, including vortices, domain walls and domains”, “control of altermagnetic configurations ranging from nanoscale vortices to domain walls and large-scale domains” etc.

However, they do not demonstrate control at the nanoscale of these textures. For this to be the case, I would expect the authors to demonstrate the reproducible nucleation and manipulation of such textures. In another paper of the group (same first author, Ref 27) they describe the generation and control of antiferromagnetic half-skyrmions. There, the control refers to the propagation of the textures with currents. Following the language of their previous paper, this work rather reports the observation and generation of textures, and not the control.

Finally, I find the conclusion and outlook quite over the top. How will these results be relevant for topologically protected Chern-insulator or Majorana states? Why are micron-scale states relevant? And how will these results break the spatial limits of ferromagnetic technologies (which exist on lengthscales an order of magnitude smaller, at least)? I do believe that the results will be impactful, but this overselling rather detracts from the results, rather than adds to it.

2. It is not clear whether there is any new physics here.

This paper combines three key aspects to achieve the imaging of altermagnetic configurations in films and heterostructures:

- The XMCD detection of the time reversal symmetry breaking providing the sign of L: Already observed in Ref 12, same authors.
- The coupling of the magnetic field to L: Already observed in Ref 12, same authors.
- The influence of the micro-patterning on the antiferromagnetic configuration: Already demonstrated in 2011 in Ref 29.

These three aspects are not new, and the authors do not provide an explanation of these phenomena, instead simply referring to the previous papers where they are discussed. What is new here, is that the authors combine the XMCD of altermagnets with X-PEEM imaging to image altermagnetic configurations: this is a very impressive and exciting result for the community, and will be key to understanding the physics of altermagnets in the future. However, there is little discussion of the physics of the presented results, and how the results relate to the altermagnetic nature of the material.

As a result, the paper comes across more as a report/ technique development based on known phenomena, with limited new physics or insight, or even discussion thereof.

Indeed, the observed domain structure is very similar to the observed configuration in antiferromagnetic hematite in Ref 24. In addition, the control of the state with microstructuring is very similar to the work of Ref 29.

The paper would benefit significantly from more discussion of the underlying physics.

For example, I would have the following questions:

- Is there any difference between the influence of microstructuring on antiferromagnets and for altermagnets?
- What is the mechanism for the “patterning” of the altermagnetic state? Are there local strain gradients in the material? What

are the "surface effects" that the authors mention?

- Can the authors provide an accessible/ intuitive explanation of the origin of the XMCD from the altermagnet, or the coupling of B to L? Both phenomena are not widely known or understood.

- What is the magnetic configuration of the domain walls and vortex cores? How large are they, are the domain boundaries smooth or abrupt, and can they be resolved by the PEEM? Does the altermagnetic nature of the material lead to some emergent effects in these textures, or is this simply an antiferromagnetic state?

- Can the authors combine with AHE measurements to "prove" the magnetic configuration corresponding to the many transport measurements that have been reported so far?

Further comments/ questions:

- The authors state that the advantage of PEEM is the combination of microscale field of view and nanoscale detection. This is an odd choice to highlight, as this is typical of almost every nanoscale microscopy technique. Surely the sensitivity of PEEM, and access to resonant scattering/ magnetic dichroism is more important?

- Typically, if you think of XMLD and XMCD, you think of a ferromagnet or a canted antiferromagnet. Can the authors rule out that the XMCD comes from canting of the antiferromagnet? In their spectroscopy paper (Ref 12), they compare high field and zero field, and see different XMCD spectra corresponding to the claimed altermagnetic order, and a net magnetic moment. However, they do not show the spectrum of their measured XMCD, which will be crucial to proving that it is of altermagnetic origin. The authors should provide such a spectrum, and explain why they believe that their signal is due to the altermagnetic order. They mention that for normal incidence XMCD PEEM such a remanent magnetisation would not lead to a signal, why not?

- Figure 2: this is not a vortex pair, but an antivortex pair. Looking at all patterned samples, it seems that with a global field and a closed surface, only anti-winding of 720 degrees can be achieved. Is there any way to obtain vortices, or are they limited to antiwinding / antivortex structures?

- The authors should take care of including scale bars and correct colour bars in figures. For example, in Fig 3, it is difficult to tell the difference between the XMCD and XMLD images from the figure.

In conclusion, the technique development to image the altermagnetic configuration, and demonstration of the response of the altermagnetic configuration to magnetic fields, is of interest to the community. However, for publication in a journal such as Nature, I would expect new insights into the physics of altermagnets, or for these technique developments to be substantial, which at this time is not the case. As a result, although these results are a very welcome advance for the field of altermagnetism, I do not believe that the manuscript is suitable for publication in Nature.

Version 1:

Reviewer comments:

Referee #1

(Remarks to the Author)

I have thoroughly reviewed the revised manuscript and the response letter, and am satisfied that the two points I raised have been adequately addressed. In my opinion, the concerns raised by other reviewers have been well resolved through additional results and revisions to the text.

I believe that the revised manuscript meets the high standards required for publication in Nature and will appeal to a wide scientific audience.

Referee #2

(Remarks to the Author)

I appreciate the efforts by the authors to revise their MS based on the comments by the reviewers.

I read the entire MS again and I am still not convinced that the major issue, ie. overselling their interesting findings has been addressed appropriately.

I strongly recommend to try one more time to focus on the actual findings of this work, which are interesting on their own and lower the language to avoid the notion of overselling it.

I also feel that the broad brush of connection fundamental aspects with applications is in most cases more confusing than helpful.

For example: "For spintronics, altermagnets can merge favorable characteristics of conventional ferromagnets and antiferromagnets, considered for a century as mutually exclusive." This is a true statement, but there is more than spintronics, and the basic nature of this work does not need the connection to potential applications.

At this stage, I would recommend to focus on the following aspects

- the abstract is still confusing. I understand that the authors try to justify their work by relating to high level aspects, but what comes across is a mix of basic principles and phenomena with experimental efforts and technological applications in a way that is hard to follow unless the reader knows already what the authors want to say. It does not really summarize the actual findings of this work, but tries to create a story to "catch it all", which still gives the impression of "overselling".
- Given that the actual MS is rather short, I highly recommend to move parts from the SI section into the main text. There are two instances in the main text, which I suggest to consider
 - p4 spectral dependence of the data. This is important information and should be included in the Main part
 - p6 stabilization of isolated Bloch-type vortices: this is a major result of this work and needs to be explained in the main text.

Before I am able to make recommendation, I would like to see those concerns being addressed.

Referee #3

(Remarks to the Author)

Review of Amin et al., "Altermagnetism imaged and controlled down to the nanoscale"

In my opinion, the authors have generally answered the concerns of the referees, and have satisfied many of my concerns, both with added explanations, and additional data presented in the SI.

However, while the reply is very informative, the authors have not made significant changes to the manuscript itself – in particular, many of the physical insights that they now have shown in the referee reply / SI are missing from the manuscript. The figures have barely changed. They should properly revise the manuscript to improve the quality and include additional physical insights (including spectra, lengthscales, domain wall profiles, and the nucleation of vortices in triangles). Once they have done this, I would be willing to consider it for publication in Nature.

Specific comments:

- I appreciate that the authors have slightly modified their abstract, intro and conclusions to address the previous concerns. I feel it is much more representative of the work.
- The discussion and extra data describing relevant lengthscales is appreciated. Normally such lengthscales are defined by a competition of competing factors, can they provide some discussion of this? How does it relate to the competition between the magnetoelastic and magnetocrystalline anisotropy?
- They now attribute the shape anisotropy to magnetoelastic interactions. However, no mapping of the strain in the sample is given. This should be possible to calculate or simulate. Can they do this, to provide additional insight / evidence as to the mechanism of the shape anisotropy?
- I appreciate the authors including the discussion of the origin of the XMCD. They could still make this discussion a little more accessible to non-experts.
- I appreciate the authors adding a spectrum to distinguish the XMCD arising from m and L. Is this spectrum measured in the PEEM on the same sample? The corresponding XAS spectrum should be plotted as well, to show the relative energy of the XMCD peaks with respect to the absorption edge.
- The added analysis of the structure, showing the width of the domain walls is appreciated. Should be included in the main figures and text.
- The patterning / nucleation of vortices with triangle shapes, and antivortices with hexagonal shapes, is very nice. Should be added to the main text. Can the authors comment on why the symmetry of the hexagon or triangle leads to different topological textures? I.e. what is the mechanism for this?
- Control: this seems to be a disagreement about wording: I am still not convinced. I see the micropatterning as a nice way to form particular structures/ states, e.g. vortices domain walls, and so they have control of the configuration via shape anisotropy, analogous to simple ferromagnets. But they do not have control of altermagnetic textures such as vortices and domain walls, as they cannot manipulate them.
- "The X-ray dichroism vector mapping used here can be combined with other imaging techniques, such as X-ray laminography, potentially offering depth sensitivity and even higher spatial resolution [35]." – laminography is a similar technique to transmission tomography (simply a slightly different geometry). While it is true this could provide depth resolution, it does not provide higher spatial resolution. Perhaps the authors mean holography or ptychography, which offer diffraction limited spatial resolution?

In conclusion, while I appreciate the efforts of the authors to reply to my previous concerns, and to provide associated physical insights, I do not believe the manuscript has improved enough to be suitable for publication in Nature. If the authors modify the manuscript to reflect their previous reply, and answer the concerns above, I would be open to considering it for publication.

Version 2:

Reviewer comments:

Referee #2

(Remarks to the Author)
I have no further comments.
I recommend to publish as is.

Referee #3

(Remarks to the Author)
The authors have implemented changes to the manuscript following my comments in the previous round of review.

In general, the quality of the manuscript is now significantly improved, and I would be happy to support its publication.

One small note: I see that the authors do not intend to make their data open access. I find this surprising and quite disappointing, as it goes against the current trends in open data. For such a new topic, and high impact result, open data is particularly important. I would encourage making the raw data available on an online repository.

Referees' comments:

Referee #1 (Remarks to the Author):

This manuscript presents spatially-resolved, full-vector imaging of altermagnetic domains in α -MnTe by combining XMLD-PEEM with XMCD-PEEM. As well explained in the manuscript, the Néel order of collinear antiferromagnets does not exhibit XMCD contrast due to time reversal symmetry. In contrast, the order parameter of altermagnets, which breaks time reversal symmetry, allows for the detection and full-vector imaging of the direction of the order parameter. This key distinction between collinear antiferromagnets and altermagnets has enabled the authors to image various spin textures, such as vortices, anti-vortices, and their pairs. Additionally, the authors have demonstrated clear domain wall formation and achieved a single domain state through field cooling in unfilled hexagon samples.

An important feature of altermagnets is their antiparallel alignment of atomic moments in real space, akin to collinear antiferromagnets, combined with spin-split bands in momentum space, characteristic of ferromagnets. This unique combination offers both advantages of antiferromagnets (i.e., no net magnetic moment and THz dynamics) and ferromagnets (i.e., directional spin polarization), which was mutually exclusive before. As a result, the altermagnetism attracts significant interest across magnetism, spintronics, and broader fields of condensed matter physics. Previous studies have identified fundamental characteristics of altermagnets such as the anomalous Hall effect, spin splitter torque, and spin-split bands. These features critically depend on altermagnetic domain structures, as different domains produce different signals with varying signs, leading to signal cancellation in multi-domain structures. Thus, investigating altermagnetic domain structures and developing methods to achieve single-domain states are the most important next steps for altermagnet studies. This work convincingly demonstrates imaging of altermagnetic domain structures and, importantly, methods to achieve domain wall and single domain states, providing a significant advancement towards future research on altermagnetic dynamics and potential applications. Moreover, the manuscript is well-written and organized. In this respect, I support publication of this manuscript in Nature after the authors address the following comments in revision:

1. Please provide more details on the influence of magnetoelastic forces and surface anisotropy in aligning the L-vector with respect to a patterned edge, as stated. Understanding this mechanism is crucial for explaining the difference in domain structures between smaller and larger hexagon samples (Fig. 2), as well as the formation of the single domain state (Fig. 3). Additionally, please clarify whether the orientation of altermagnetic domains is determined by the edge direction rather than the magnetic easy axis of α -MnTe. What implications arise if the edge direction differs from the magnetic easy axis?

Answer

We thank the referee for their valuable comments and questions. To provide further insight into the competing edge anisotropy and intrinsic magnetocrystalline anisotropy in our α -MnTe structures, **we have added three sections to the supplementary material (section S6, S7, and S8).**

In **supplementary section S6** we present a large open-space region in proximity to a patterned edge, from which we measure an effective length-scale ($1.7\mu\text{m}$) of the edge anisotropy for both hard and easy axes edges. **Supplementary section S7** shows an unfilled hexagon, with $4\mu\text{m}$ bar widths. In the virgin state and field-cooled state of this microstructure, a multidomain state is observed at the centre of each $4\mu\text{m}$ bar size, again consistent with a relaxation of the edge anisotropy over a length-scale of $\sim 1.7\mu\text{m}$. This, therefore, provides an upper limit for the formation of single domain states in our patterned bars.

The referee also highlights an important point about the alignment of the patterned edge direction with the magnetic easy axes and the implications of aligning the patterned edge away from these axes. We have determined that the edge anisotropy (over the effective $\sim 1.7\mu\text{m}$ length-scale) dominates the domain

formation in our α -MnTe films, rather than the intrinsic magnetocrystalline anisotropy. This is the case whether the edges are patterned along the $\langle 1-100 \rangle$ easy or $\langle 2-1-10 \rangle$ hard magnetic axes. This is verified in **supplementary section S8**, in which we compare domain formation in easy-axes and hard-axes unfilled hexagons with $2\mu\text{m}$ bar width. Alignment with the bar edges in both cases is confirmed using XMLD-PEEM. We show that, consistent with the predicted symmetry of the Néel-vector induced XMCD, we detect vanishing XMCD-PEEM contrast for the domains parallel to the $\langle 2-1-10 \rangle$ axes.

Similar edge-induced magnetic anisotropy has been observed in several epitaxial, compensated magnetic materials, such as the collinear traditional antiferromagnets, LaFeO_3 ⁽¹⁾, NiO ⁽²⁾, CuMnAs ⁽³⁾, and Mn_2Au ⁽³⁾. It is understood, from experimental studies of these materials and combined with empirical models, that magnetoelastic interactions due to spontaneous strains induced by the domain configuration, and elastic strain due to the substrate, play a dominant role in defining the edge anisotropy.

⁽¹⁾ E. Folven et al. *PRB* 84.22 (2011)

⁽²⁾ H. Meer et al. *PRB* 106, 094430 (2022)

⁽³⁾ S. Reimers et al. *Phys. Rev. Appl.* 21, 064030 (2024)

2. I assume that the α -MnTe imaged in this study is a single crystal, both in-plane and along the thickness direction. This is important as the order parameter of an altermagnet is magnetic multipole, which results from the product of electric multipole with spin, forming a ferroic order that breaks time reversal symmetry and enables XMCD imaging. The orientation of the electric multipole directly correlates with the crystallographic orientation, limiting the size of altermagnetic domains below the size of crystallographic grains. For practical applications, using sputtering (unlike the MBE method employed in this work) is common but introduces many small grains with random orientations, complicating the realization of well-defined domain walls and single domain states. While this does not detract from the novelty and significance of the current work, a discussion on this issue would motivate the reader to investigate a way of achieving well-defined domain wall and single domain states in sputtered altermagnets, which is of crucial importance for applications.

Answer

The referee has raised an important question about the viability of growing α -MnTe films using techniques other than molecular beam epitaxy (MBE), such as sputtering, which may be applicable to wider-scale applications. Our X-ray diffraction data, presented in supplementary material **section S1**, confirms that our MBE grown α -MnTe forms high quality, single crystalline films, with well-defined sixfold (hexagonal) symmetry in the ab plane. This coherent orientation is necessary to produce altermagnetic single domain states and smoothly varying textures, such as domain walls and vortices.

Though epitaxial growth is readily achieved using MBE, we note it may also be possible using magnetron sputtering, as has recently been reported in an ARPES study of altermagnetic CrSb [S. Reimers et al., *Nature Commun.* 15, 2116 (2024)].

We have included this point in the methods section of the main manuscript:

While MBE is a standard technique for growing epitaxial thin films, we note that sputtering has also been used to grow high quality altermagnets, such as CrSb [17].

Referee #2 (Remarks to the Author):

A. Summary of the key results

The MS by Amin et al reports an experimental study of a thin film of α -MnTe (0001) which is supposed to be an altermagnetic material. The sample was epitaxially grown on an InP(111) substrate and the signature features of an altermagnet was validated using XMCD and XMLD with an X-PEEM.

The topic of altermagnetism is very timely as this new class of magnetic materials should change the entire paradigm of how to look at magnetism. Altermagnetism (AM) combines features of antiferromagnetism (AF), i.e. no net magnetic moment with features from ferromagnetism (FM), i.e. spin ordering in q -space. Since FM and AF exhibit different T-reversal symmetry, this is scientifically a fascinating topic and might lead to unprecedented opportunities with future magnetic devices.

The approach by this study is based on a combination of the XMCD effect, which is sensitive to FM, i.e. magnetization $\langle M \rangle$, and with XMLD, which is known to be primarily sensitive to AF ordering, i.e. $\langle M^2 \rangle$. The microscopic magnetism of the MnTe film is addressed by using both dichroism effects as contrast mechanism in a spatially resolving state-of-the-art X-PEEM instrument at MAX-IV, where the alignment of the x-ray beam and the sample orientation can be varied.

The authors also demonstrate the modulation of the AM state via external fields and temperature in a micropatterned system.

It is important to note that the authors have published a preliminary magnetic X-ray spectroscopy study with this material (not clear whether it was the exact sample), where they observed a crystal orientation dependent XMCD signal, which is an indicator for the k -dependent spin polarization in an altermagnetic system.

B. Originality and significance: if not novel, please include reference

The work presented is highly novel and of great interest to a broad audience, therefore considering it for a publication in Nature is justified. The fact that the first milestone, i.e. the non-spatially resolved XMCD spectroscopy study was published previously in *PRL*, is not necessarily a weakness, although it could have been combined into a single high profile publication

Answer

We thank the referee for their comments. Our published *PRL* paper on non-spatially resolved XMCD spectroscopy is intended to give a comprehensive and rigorous (mainly theoretical) description of the origin of circular magnetic dichroism in our altermagnetic material, α -MnTe. This provides a basis for us to embark on this detailed spatially resolved experimental work. We agree with the referee that the present work will be of interest to the broad audience of Nature, while the underpinning work we felt was more appropriate to the more specialised focus of *PRL*.

C. Data & methodology: validity of approach, quality of data, quality of presentation

The data quality is high, but I have a concern regarding the validity of the approach. The depth resolution of X-PEEM is limited and therefore it is unclear whether the surface properties impact the conclusion about the altermagnetic phase. Also, most of the theory work on AM materials seems to be bulk focused and it would be interesting to know, whether the thickness of the sample and the above mentioned depth limited sensitivity of X-PEEM could have an impact to the conclusion. In other words, can we expect that any bulk crystal that shows AM behavior can equally be fabricated in thin film form and still maintain the AM character. For the technological applications of AM materials, this would be of significant importance.

Answer

We thank the referee for raising this point. **We have added a new section (S3) to the supplementary material to further validate our approach.** In section S3, we show that the XMCD-PEEM contrast, in the open-space region presented in Fig.1 of the main text, reverses at X-ray energies across the Mn L_2 resonance edge. This follows the uniquely altermagnetic bulk-like behaviour as determined from ab initio calculations and non-spatially resolved X-ray spectra (details provided in Ref.12⁽¹⁾).

The probing depth of the X-PEEM is determined by the electron escape depth which is ~5nm. This is a significant fraction of the total 30nm thickness of our α -MnTe films. While we cannot rule out different behaviour at the bottom MnTe/InP interface, we do not expect to see qualitative changes in the magnetic domain structure over this length-scale. By comparison, the observed lateral 180° domain walls in our samples have a characteristic width of ~100nm, i.e. the in-plane spin rotation over a considerably larger length-scale compared with than the film thickness. We also emphasize that the symmetries we observe in our 30nm films of α -MnTe follow the expected symmetries of the bulk altermagnetic phase.

The question of whether any bulk altermagnetic crystal can be expected to maintain its altermagnetic character in thin film form is a very broad one. In the case of MnTe, spontaneous anomalous Hall effects have been observed in both bulk crystals and thin films^(2,3). Altermagnetic characteristics (e.g. spin splitting torques, band splitting or anomalous Hall effects) have elsewhere been reported for thin films of RuO₂, CrSb and Mn₅Si₃ (see, e.g. Refs.5⁽⁴⁾,6⁽⁵⁾,8⁽⁶⁾,10⁽⁷⁾,13⁽⁸⁾,17⁽⁹⁾). However, as described in our response to Review #1's Q2, it is important to maintain a coherent crystallographic orientation, while epitaxial strain and defects introduced during thin film growth may affect their magnetic properties.

(1) A. Hariki et al. *PRL* 132.17 (2024)

(2) K. P. Kluczyk et al. *arXiv:2310.09134*

(3) R. D. Gonzalez Betancourt et al. *PRL* 130.3 (2023)

(4) L. Šmejkal et al. *Nature Reviews Materials* 7.6 (2022)

(5) H. Reichlová et al. *arXiv preprint arXiv:2012.15651* (2020)

(6) T. Tschirner et al. *APL Materials* 11.10 (2023)

(7) M. Wang et al. *Nature Communications* 14.1 (2023)

(8) O. Fedchenko et al. *Science advances* 10.5 (2024)

(9) S. Reimers et al. *Nature Communications* 15.1 (2024)

Regarding the presentation itself, I have several comments that need to be addressed before I can recommend it for publication in any peer-reviewed journal

- The abstract is confusing and does not reflect the actual work. It needs to be rewritten

Answer

In accordance with the referee's suggestion, we have made changes to the abstract. It now reads as follows:

Nanoscale detection and control of the magnetic order underpins a broad spectrum of fundamental research and practical device applications. The key principle involved is the breaking of time-reversal (T) symmetry, which in ferromagnets is generated by an internal magnetization. However, the presence of a net-magnetization also imposes severe limitations on compatibility with other prominent phases ranging from superconductors to topological insulators, as well as on spintronic device scalability. Recently, altermagnetism has been proposed as a solution to this restriction, since it shares the enabling T-symmetry breaking characteristic of ferromagnetism, combined with the antiferromagnetic-like vanishing net-magnetization [1–4]. To date, altermagnetic ordering has been inferred from spatially averaged probes [4–19]. Here, we demonstrate nanoscale imaging of altermagnetic states ranging from ~ 100 nm scale vortices and domain walls to ~10 μ m scale single-domain states in MnTe [2, 7, 9, 14–16, 18, 20, 21]. We combine the T-symmetry breaking sensitivity of X-ray magnetic circular dichroism [12] with magnetic linear dichroism and photoemission electron microscopy, to achieve detailed imaging of the local altermagnetic ordering vector. A rich variety of spin configurations can be imposed using microstructure patterning and thermal cycling in magnetic fields. The demonstrated detection and controlled formation of altermagnetic spin configurations paves the way for experimental studies across the theoretically predicted broad research landscape of altermagnetism, ranging from unconventional non-relativistic and relativistic spin-polarization phenomena, the interplay of altermagnetism with superconducting and topological phases, to highly scalable digital and neuromorphic spintronic devices [3, 14, 22-24].

- There are several instances of “generic phrases”, which I recommend to reconsider. Examples are “abundance of robust altermagnetism” (is this scientifically justified?), “offering the optimal combination of microscale field of view and nanoscale detection limit”. I would question this

statement entirely. There are several other examples in the text that I hope that authors will be able to either provide clarity or rephrase.

Answer

We thank the referee for their suggestion. We have revised the wording of the highlighted phrases in the introduction of the main manuscript. Paragraphs 1 and 2 of the main manuscript now read as follows:

For condensed matter physics, the d-wave (or higher even-parity wave) spin-polarization order in altermagnets represents the sought-after, but for many decades elusive, counterpart in magnetism of the unconventional d-wave order parameter in high-T_c superconductivity [3]. For spintronics, altermagnets can merge favorable characteristics of conventional ferromagnets and antiferromagnets, considered for a century as mutually exclusive [3]. They can combine strong spin-current effects, which underpins reading and writing functionalities in commercial ferromagnetic memory bits, with vanishing net magnetization, enabling demonstrations of high spatial, temporal and energy scalability in experimental antiferromagnetic bits insensitive to external magnetic field perturbations. These examples, as well as the predicted abundance of altermagnetic materials, ranging from insulators and semiconductors to metals and superconductors, illustrate the expected broad impact of this new field on modern science and technology [3].

To date, however, the unconventional properties of altermagnets have been experimentally detected using spatially-averaging electronic transport [4–11] or spectroscopy probes [12–19]. In our work, we report mapping of the altermagnetic order-vector and demonstrate the controlled formation, from nano- to microscale, of a rich landscape of altermagnetic textures, including vortices, domain walls, and domains. We employ polarized X-ray photoemission electron microscopy (PEEM), which is a powerful tool in magnetism, allowing for, in addition to element specificity and magnetic sensitivity, concurrent full-field real-space imaging at the microscale with nanoscale resolution.

- P4: “full-vector mapping” It is important to point out that the 3D vector magnetization is only obtained (if at all) in the detectable depth of X-PEEM. And, even there, I doubt that the sensitivity is depth-independent, which means that other techniques, such as X-ray laminography, where the same contrast mechanisms can be employed, might provide a more accurate mapping.

Answer

We thank the referee for raising this important point and acknowledge our use of the phrase ‘full vector mapping’ to be misleading in this context. Our aim was to highlight the sensitivity of our combined XMLD and XMCD measurements to the Néel vector direction, in contrast to solely using XMLD (which determines the axis of the Néel vector and not the sign), or XMCD (which is insensitive to the Néel vector in conventional collinear antiferromagnets). However, as we do not provide depth dependence, we have replaced the phrase ‘full vector mapping’ with ‘vector mapping’ throughout the manuscript.

To broaden the discussion to alternative imaging techniques, we have added, in the concluding paragraph of the main manuscript, the following reference to X-ray laminography, which, as the referee rightly comments, may offer enhanced sensitivity and depth-dependence:

The X-ray dichroism-based vector mapping used here is also compatible with other imaging techniques such as X-ray laminography, potentially offering depth sensitivity and even higher spatial resolution [Witte, Katharina, et al. Nano letters 20.2 (2020): 1305-1314].

- Figure captions/ labelling: The authors need to provide scale bars and statements about the spatial resolution in the obtained images. The color wheel in Fig 1b is too small, it should be enlarged, since this is the essential information to follow the conclusions.

Answer

We thank the referee for bringing this to our attention. We have added spatial scale bars to all X-PEEM images in Figs.1-3 and enlarged the colour wheel schematic in Fig.1b.

- Fig 1c: Is this really all magnetic contrast? What is the sample roughness?

Answer

The XMCD-PEEM image presented in Fig.1c is almost entirely magnetic contrast. We calculate this image as the asymmetry between images taken with opposite X-ray circular polarisation helicity. Since structural features are insensitive to the X-ray helicity, only the magnetic contrast remains.

To clarify this, **we have added two new sections (S3 and S4) to the supplementary material to verify that our contrast is of magnetic origin.** In **section S3** we show contrast reversal in XMCD-PEEM images measured at three energies across the Mn L_2 resonance edge. The sign of the XMCD contrast follows the uniquely alternating magnetic behaviour expected as determined from non-spatially resolved XMCD spectra. In **section S4** we show a region of open-space imaged at low temperature, where the XMCD contrast is clear, and at high temperature, where the XMCD contrast vanishes. This confirms that there is negligible structural contrast in our XMCD-PEEM images.

- Fig 1f: what are the black rims surrounding the colored areas? The two circles indicating the V-AV pair are confusing. I suggest to consider a closer zoom-in for those regions.

Answer

We thank the referee for their question and suggestion. The choice of our colour map for the vector map images in Figs.1-3 is intended to highlight the spin axes of magnetic domains parallel to the $\langle 1-100 \rangle$ easy axes. Dark colour between the boundaries of coloured domains correspond to 60° domain walls where the spin axis varies through the $\langle 2-1-10 \rangle$ hard axes. This follows the convention of previously reported XMLD vector map studies⁽¹⁾. In **section S12 of the supplementary material**, we show the smooth Néel vector variation across 60° domain walls (dark boundaries in the colour map) and determine the average domain wall thickness to be ~ 70 nm.

We have changed the size of Fig.1c to help with identification of the vortex-antivortex pair. The coloured circles highlight the positions of the vortex cores, while not compromising the interpretation of the surrounding spin variation.

⁽¹⁾ H. Jani, et al. *Nature* 590.7844 (2021)

- Fig 2: what are the gray areas surrounding the magnetic domains?

Answer

In Fig.2 the ~ 200 nm grey border of the hexagon structures are where our patterning procedure (ion milling) forms trapezoid edges. While these boundaries are visible in our XMCD-PEEM images, there is no resolvable magnetic contrast. An outline of the hexagon edges has been included in our vector colour maps (Figs.2c,e,g,i) to show the full scale of the patterned hexagons, consistent with the raw XMCD-PEEM images (Figs.2b,d,f,h).

- Fig 3: Micropatterning the AM system: how relevant are the length scales of the microstructure

Answer

We thank the referee for their question. **We have added three sections to the supplementary material (S6, S7, and S8)** to elucidate the relevance of the length-scale of our patterned microstructures. We directly determine, from XMLD- and XMCD-PEEM images of an open-space region in proximity to a patterned edge, the length-scale ($\sim 1.7\mu\text{m}$) that the edge anisotropy is effective over (see **section S6**). In **section S7**, we show that an unfilled hexagon, with $4\mu\text{m}$ bar widths, in the virgin state and field-cooled state, is the upper limit microstructure size to forming single domain states. Finally, in **section S8**, we show that an unfilled hexagon, with $2\mu\text{m}$ wide bars patterned along the $\langle 2-1-10 \rangle$ hard-axes, stabilizes single domains parallel to the bar direction, from which we infer that the edge anisotropy, on this length-scale, is a more significant energy contribution than the intrinsic magnetocrystalline anisotropy.

D. Appropriate use of statistics and treatment of uncertainties

The analysis of the data is state-of-art, however, as mentioned above a critical assessment of the limitations of the X-PEEM technique as such and mentioning potential alternatives is missing

Answer

We thank the referee for their comment. We refer to our answer, to the above related question, in which we address this point.

E. Conclusions: robustness, validity, reliability

The conclusions drawn are consistent within the limits of the experimental sensitivities of X-PEEM.

F. Suggested improvements: experiments, data for possible revision

See detailed comments under C:

G. References: appropriate credit to previous work?

The list of references is appropriate, and the authors are commended for including even all the arXiv papers in this rapidly evolving field

H. Clarity and context: lucidity of abstract/summary, appropriateness of abstract, introduction and conclusions

See detailed comments under C:

In summary, I find this work highly interesting, very timely and relevant to the current discussion in the magnetism community about AM as a new class of magnetic materials. Questions that need to be addressed are the general validity of AM in thin films, which would be most relevant to technological applications. The discussion about topological features, ie. the V-AV pair could be expanded.

Before I can recommend it for publication, the authors need to address my comments above.

Answer

We thank the referee for their valuable questions and suggested modifications. We feel the changes made to the main manuscript, as well as the additional supplementary material, have strengthened the delivery of our results and we hope they fully address the referee's comments.

Referee #3 (Remarks to the Author):

Amin et al present the imaging of the magnetic configuration of the altermagnetic material MnTe. Through the combination of the X-ray magnetic circular and linear dichroism, which probe time-reversal symmetry breaking, and the Neel orientation, respectively, they map the altermagnetic vector field in a 30 nm thick MnTe film, revealing the presence of vortices, antivortices, and domain wall textures. They demonstrate that by field cooling, they can alter the magnetic configuration. In addition, they go on to demonstrate that by patterning the film into micro-hexagons and microwires, they can achieve local ordering of the Neel vector.

Altermagnetism is currently a very “hot topic” in the condensed matter physics community, with many theory papers, and a growing number of experimental works being published in recent years. This work represents the first spatially resolved measurements of the magnetic configuration of an altermagnet, providing insight into the types of textures that form, as well as the influence that microstructuring can have on the magnetic configuration. The work is very timely, and I’m sure will be of interest to many in the community.

However, there are two main reasons why I believe that this manuscript is not suitable for publication in Nature:

1. I find the results are oversold.

The title “Altermagnetism imaged and controlled down to the nanoscale” does not reflect the achievements of the paper. Yes, the manuscript presents the imaging of altermagnetic domains at the nanoscale. However, the title is misleading in that it implies that the magnetic configuration can be controlled at the nanoscale, which is not the case. The authors show that they can create micro-metre-sized domain structures with micropatterning. However, it is not the case that the state can be precisely or locally *controlled* at the nanoscale, but rather that they observe the influence of surface effects and magnetoelastic effects on the local magnetic configuration.

Answer

We thank the referee for their positive assessment of the quality and timeliness of our manuscript. Regarding the general concerns raised we would like to highlight that, in our study, we leverage a combination of pattern-induced edge anisotropy and magnetic field-cooling to deterministically nucleate domain states ranging from micron-sized single domains to nanoscale domain walls and vortices. We control the sign of the Néel vector, by choice of externally applied field during field-cooling, which we determine from our XMLD- and XMCD-PEEM mapping technique. In a newly added supplementary material section, we demonstrate that field-cooled triangle microstructures stabilize isolated Bloch-type vortices (~100nm radius) centrally located with nanoscale precision, whose vorticities are controlled by the geometric orientation. This is a level of control of a nanoscale texture that is unprecedented in a compensated magnet.

In terms of the field of altermagnetism itself, there are many claims in the first paragraph of the main text, and the authors give a misleading impression that the field is mature and well established. However, only one perspective article is cited. For example, they claim there is an “abundance of robust altermagnetism among materials ranging”. I would disagree with this: the field is still very young, and while very exciting, is still developing. The authors should soften their claims in the introduction of the manuscript, and provide a more balanced discussion.

Answer

We thank the referee for their comments. We have accordingly clarified the phrase ‘abundance of robust altermagnetism’. The first paragraph of the main manuscript now reads:

For condensed matter physics, the d-wave (or higher even-parity wave) spin-polarization order in altermagnets represents the sought-after, but for many decades elusive, counterpart in magnetism of the unconventional d-wave order parameter in high-Tc superconductivity [3]. For spintronics, altermagnets can merge favorable characteristics of conventional ferro-

magnets and antiferromagnets, considered for a century as mutually exclusive [3]. They can combine strong spin-current effects, which underpins reading and writing functionalities in commercial ferromagnetic memory bits, with vanishing net magnetization, enabling demonstrations of high spatial, temporal and energy scalability in experimental antiferromagnetic bits insensitive to external magnetic field perturbations. These examples, as well as the predicted abundance of altermagnetic materials, ranging from insulators and semiconductors to metals and superconductors, illustrate the expected broad impact of this new field on modern science and technology [3].

The authors mention multiple times that they achieve a “full-vector mapping”. However, they only map the vector in the plane. A full vector mapping would be three dimensional.

Answer

We thank the referee for raising this important point and acknowledge our use of the phrase ‘full vector mapping’ to be misleading in this context. Our aim was to highlight the sensitivity of our combined XMLD and XMCD measurements to the Néel vector direction, in contrast to solely using XMLD (which determines the axis of the Néel vector and not the sign), or XMCD (which is insensitive to the Néel vector in conventional collinear antiferromagnets). We also emphasize that, in our α -MnTe films, the Néel vector is contained within the *ab* plane, with the exception of vortex core spins, which are below the resolution of the PEEM to be detectable. As such, we believe our 2-dimensional vector maps to be a true representation of the Néel vector direction. However, as we do not provide explicit out-of-plane sensitivity in our mapping, we have replaced the phrase ‘full vector mapping’ with ‘vector mapping’ throughout the manuscript.

The authors mention many times “control of configurations”: e.g. “demonstrate the control, from nano- to microscale, of a rich landscape of altermagnetic textures, including vortices, domain walls and domains”, “control of altermagnetic configurations ranging from nanoscale vortices to domain walls and large-scale domains” etc.

However, they do not demonstrate control at the nanoscale of these textures. For this to be the case, I would expect the authors to demonstrate the reproducible nucleation and manipulation of such textures. In another paper of the group (same first author, Ref 27) they describe the generation and control of antiferromagnetic half-skyrmions. There, the control refers to the propagation of the textures with currents. Following the language of their previous paper, this work rather reports the observation and generation of textures, and not the control.

Answer

We use the term control to refer to the deterministic formation of micron-size domain states and nanoscale textures in our patterned microstructures. We demonstrate the ability to select the nucleation and chirality of ~ 100 nm vortices, or the formation of large single domain states by patterning and thermal cycling in an external magnetic field. Reversing the direction of the field during thermal cycling allows us to control the Néel vector direction. This level of Néel vector manipulation, to the point of determining the character of nano- and microscale textures, is unprecedented in a compensated magnetic material.

Finally, I find the conclusion and outlook quite over the top. How will these results be relevant for topologically protected Chern-insulator or Majorana states? Why are micron-scale states relevant? And how will these results break the spatial limits of ferromagnetic technologies (which exist on lengthscales an order of magnitude smaller, at least)? I do believe that the results will be impactful, but this overselling rather detracts from the results, rather than adds to it.

Answer

We thank the referee for raising this point. We have made changes to the concluding paragraph of the main manuscript in accordance with the referee’s suggestion:

*The vector imaging and controlled formation of altermagnetic configurations ranging from nanoscale vortices and domain walls to microscale domains, demonstrated in this work, has broad science and technology implications. It is the basis on which the new experimental field can develop, leveraging the *T*-symmetry breaking phenomenology, vanishing magneti-*

zation, ultrafast dynamics, and predicted compatibility of the altermagnetic order with the full range of conduction types from insulators to superconductors [3]. The X-ray dichroism vector mapping used here can be combined with other imaging techniques, such as X-ray laminography, potentially offering depth sensitivity and even higher spatial resolution [35]. The ability to image and control the formation of microscale single-domain states will be highly relevant in the experimental research of fundamental electronic-structure properties of altermagnets, including the predicted unconventional non-relativistic and relativistic spin polarization and topological phenomena, or interplay with other order parameters such as superconductivity [3, 14, 22–24]. Similarly, the controlled spatial uniformity of the altermagnetic states is an important step for the experimental research of digital spintronic devices. Multidomain states with spatially varying magnetic configurations represent a complementary area which can leverage the unique phenomenology of altermagnets in the research of topological skyrmions, merons and other magnetic textures, and in the related field of neuromorphic spintronic devices. Our demonstration of the vector mapping and controlled formation of the altermagnetic textures opens this experimental research front.

2. It is not clear whether there is any new physics here.

This paper combines three key aspects to achieve the imaging of altermagnetic configurations in films and heterostructures:

- The XMCD detection of the time reversal symmetry breaking providing the sign of L: Already observed in Ref 12, same authors.**
- The coupling of the magnetic field to L: Already observed in Ref 12, same authors.**
- The influence of the micro-patterning on the antiferromagnetic configuration: Already demonstrated in 2011 in Ref 29.**

These three aspects are not new, and the authors do not provide an explanation of these phenomena, instead simply referring to the previous papers where they are discussed. What is new here, is that the authors combine the XMCD of altermagnets with X-PEEM imaging to image altermagnetic configurations: this is a very impressive and exciting result for the community, and will be key to understanding the physics of altermagnets in the future. However, there is little discussion of the physics of the presented results, and how the results relate to the altermagnetic nature of the material.

As a result, the paper comes across more as a report/ technique development based on known phenomena, with limited new physics or insight, or even discussion thereof.

Indeed, the observed domain structure is very similar to the observed configuration in antiferromagnetic hematite in Ref 24. In addition, the control of the state with microstructuring is very similar to the work of Ref 29.

Answer

We thank the referee for highlighting the excitement and impact of our work. We strongly agree that this work will provide the basis to develop future understanding of fundamental altermagnetic phenomena, as well as being a significant step towards realising practical altermagnetic devices. However, we acknowledge the concerns raised regarding the new physics or insight. We highlight the points that we believe are novel:

We build upon the theoretical predictions and preliminary non-spatially resolved XMCD measurements, introduced in Ref.12⁽¹⁾, to spatially map the Néel vector in our α -MnTe samples. This kind of mapping is completely novel in a compensated magnetic system and provides invaluable insight into the equilibrium domain behaviour of our material. While patterning induced anisotropies in antiferromagnets were presented in Ref.29⁽²⁾, this was not a known or expected phenomenon in an altermagnetic system. By combining micro-patterning with field cooling, we demonstrate unprecedented control over the nucleation of single domains, domain walls, and vortices with predefined chirality, confirmed by our novel vector imaging, and leveraged by the altermagnetic symmetries of α -MnTe. This is a very important enabler for further

experimental fundamental research in multiple directions. The ability to preset such well-defined magnetic states on a range of length-scales answers a major outstanding question in the field of altermagnetism.

It should be emphasized that this is the first such real-space imaging and control of the Néel vector *direction* in a compensated magnet. The altermagnetic nature of the material directly enables the results, both imaging and control, thanks to the unique T-symmetry breaking of the electronic structure and responses of altermagnets. Unlike non-collinear magnets, altermagnets have one Néel vector defined as the difference between the two opposite sublattice magnetizations, and unlike conventional collinear antiferromagnets, have the T-odd responses enabling the detection of the sign of the Néel vector.

⁽¹⁾ A. Hariki, et al. *PRL* 132.17 (2024)

⁽²⁾ E. Folven et al. *PRB* 84.22 (2011)

The paper would benefit significantly from more discussion of the underlying physics.

For example, I would have the following questions:

- Is there any difference between the influence of microstructuring on antiferromagnets and for altermagnets?

Answer

We have added a significant amount of new supplementary data to elucidate the influence of microstructuring on the magnetic domain structure in our α -MnTe films. This shows the effect of patterning along different crystalline axes (**section S6**), the length-scale over which the edge anisotropy operates (**sections S7 and S8**), and the generation of isolated vortices in triangular microstructures (**section S9**). A key difference compared to conventional antiferromagnets is that, in our system, thermal cycling in a magnetic field facilitates the alignment of the direction of the Néel vector with respect to the patterned edge, rather than just the axis, demonstrating unprecedented control over the nucleation of single domains, domain walls and vortices with predefined chirality.

- What is the mechanism for the “patterning” of the altermagnetic state? Are there local strain gradients in the material? What are the “surface effects” that the authors mention?

Answer

Our experiments suggest the effect to have the same origin as in other compensated magnetic systems. Edge-induced magnetic anisotropy has been observed in several epitaxial, compensated magnetic materials, such as the collinear traditional antiferromagnets, LaFeO₃⁽¹⁾, NiO⁽²⁾, CuMnAs⁽³⁾, and Mn₂Au⁽³⁾. To our knowledge, the first such demonstration in an altermagnetic. The prevailing understanding, from studies of antiferromagnetic materials and through empirical models, is that magnetoelastic interactions due to spontaneous strains induced by the domain configuration, and elastic strain (surface effects) due to the substrate, play a dominant role in defining the edge anisotropy. We recommend two recent papers for an in-depth picture of the patterning effect on the strain-induced anisotropy in compensated magnets^(2,3).

⁽¹⁾ E. Folven et al. *PRB* 84.22 (2011)

⁽²⁾ H. Meer et al. *PRB* 106, 094430 (2022)

⁽³⁾ S. Reimers et al. *Phys. Rev. Appl.* 21, 064030 (2024)

- Can the authors provide an accessible/ intuitive explanation of the origin of the XMCD from the altermagnet, or the coupling of B to L? Both phenomena are not widely known or understood.

Answer

We thank the referee for their question. We acknowledge that the explanation of the underlying physics is quite technical. The origin of the XMCD is described in detail in Ref.12⁽¹⁾. We have added the following paragraph to the main manuscript:

XMCD is the difference between absorption of right and left circularly polarized X-rays which, in analogy to the dc anomalous Hall effect (AHE), is given by the Hall vector $\mathbf{h}=(\sigma_{zy}^a, \sigma_{xz}^a, \sigma_{yx}^a)$. Here $\sigma_{ij} = -\sigma_{ji}$ are the antisymmetric components of the (frequency dependent) conductivity tensor. For \mathbf{L}

in the (0001)-plane of MnTe, \mathbf{h} points along the [0001]-axis, i.e. $\sigma_{zy}^a = \sigma_{xz}^a = 0$ and $\sigma_{yx}^a \neq 0$, with the exception of $\mathbf{L} \parallel \langle 2-1-10 \rangle$ -axes where also $\sigma_{yx}^a = 0$ by symmetry.

The coupling of \mathbf{B} to \mathbf{L} is mediated by the weak ferromagnetic moment, which aligns with the c -axis due to spin-orbit coupling. The weak moment couples to the magnetic field via the Zeeman interaction, and to the Néel vector via the spin-orbit interaction. This picture was convincingly demonstrated in a recent preprint (Ref.9⁽²⁾). We have added this reference to the relevant passage of text in the main manuscript:

In agreement with earlier spatially-averaging measurements of the anomalous Hall effect and XMCD spectra [7⁽³⁾, 12⁽¹⁾], and explained by the coupling of the external field to \mathbf{M} and of \mathbf{M} to \mathbf{L} [9⁽²⁾], this procedure results in the population of only one sign of L in each pair of the $\langle 1-100 \rangle$ easy-axis domains (see Fig. 2d,e).

(1) A. Hariki et al. *PRL* 132.17 (2024)

(2) K. P. Kluczyk et al. *arXiv:2310.09134*

(3) R. D. Gonzalez Betancourt et al. *PRL* 130.3 (2023)

- What is the magnetic configuration of the domain walls and vortex cores? How large are they, are the domain boundaries smooth or abrupt, and can they be resolved by the PEEM? Does the altermagnetic nature of the material lead to some emergent effects in these textures, or is this simply an antiferromagnetic state?

Answer

We thank the referee for their questions about the character of the nanoscale textures that we present in our α -MnTe films. **We have added three sections to the supplementary material (sections S10, S11, and S12) to directly address this point.**

In **section S10**, we show analysis of 180° domain wall profiles, measured in XMLD- and XMCD-PEEM images of our 2 μ m bar width, unfilled hexagon device. We show that the spatial resolution of the X-PEEM is sufficient in resolving the smooth spin variation across the ~100nm domain wall widths. Some phenomena emerging as a result of the altermagnetic properties in 180° domain walls has been theoretically modelled in a recent arxiv preprint [O. Gomonay et al. *arXiv:2403.10218*]. There it is predicted that altermagnetic domain walls exhibit unique magnetisation profiles, in contrast to their antiferromagnetic counterparts. However, a detailed study of these novel features requires extensive theoretical and experimental development.

In **section S11**, we present analogous analysis of the vortex core width in the open-space region (Fig.1 of the main manuscript). We show that the vortex profile has comparable width (~100nm) to the isolated 180° domain walls. In **section S12**, we present measurements of 60° domain wall profiles in open-space and determine the domain wall width to be ~70nm.

- Can the authors combine with AHE measurements to "prove" the magnetic configuration corresponding to the many transport measurements that have been reported so far?

Answer

Our measurements reveal a rich domain structure which is strongly influenced by applied magnetic fields, consistent with the spontaneous anomalous Hall effect (AHE) reported for α -MnTe in Ref.7⁽¹⁾ and Ref.9⁽²⁾. To highlight the connection between the AHE and the XMCD, we have added Hall resistivity measurements in **Section S4**, showing that both the XMCD domains and the hysteresis in the AHE are absent at the elevated temperature.

(1) R. D. Gonzalez Betancourt et al. *PRL* 130.3 (2023)

(2) K. P. Kluczyk et al. *arXiv:2310.09134*

Further comments/ questions:

- The authors state that the advantage of PEEM is the combination of microscale field of view and nanoscale detection. This is an odd choice to highlight, as this is typical of almost every nanoscale

microscopy technique. Surely the sensitivity of PEEM, and access to resonant scattering/ magnetic dichroism is more important?

Answer

We thank the referee for their comment. We have modified the sentence in the introduction section of the main manuscript to read:

We employ polarized X-ray photoemission electron microscopy (PEEM), which is a powerful tool in magnetism, allowing for, in addition to element specificity and magnetic sensitivity, concurrent full-field real-space imaging at the microscale with nanoscale resolution.

We intend to highlight the concurrent microscale, full-field imaging technique and nanoscale resolution that the X-PEEM offers. This is different to other nanoscale microscopy techniques that require rastering or image stitching to achieve microscale fields of view. We agree that the sensitivity of X-PEEM to resonant magnetic dichroism is critical, and we note that throughout the manuscript we state that we utilize XMCD/XMLD for nanoscale imaging of the magnetic order vector.

- Typically, if you think of XMLD and XMCD, you think of a ferromagnet or a canted antiferromagnet. Can the authors rule out that the XMCD comes from canting of the antiferromagnet? In their spectroscopy paper (Ref 12), they compare high field and zero field, and see different XMCD spectra corresponding to the claimed altermagnetic order, and a net magnetic moment. However, they do not show the spectrum of their measured XMCD, which will be crucial to proving that it is of altermagnetic origin. The authors should provide such a spectrum, and explain why they believe that their signal is due to the altermagnetic order. They mention that for normal incidence XMCD PEEM such a remanent magnetisation would not lead to a signal, why not?

Answer

We thank the referee for raising this important point. We have added **section S3 to the supplementary material** showing XMCD-PEEM images of the open-space area (Fig.1 of the main manuscript) taken at three energies across the Mn L_2 edge. Consistent with the unique character of the zero field non-spatially resolved altermagnetic XMCD spectrum, at the chosen energies the XMCD-PEEM contrast (obtained as the asymmetry between opposite photon helicities) is completely reversed. In the case of a ferromagnetic-like Mn L_2 XMCD signal, the spectrum only shows positive values and the XMCD-PEEM contrast would not reverse. This shows conclusively that the spectral character of these layers mirrors the sister layers studied in the non-spatially resolved XMCD in Ref.12 [A. Hariki et al. *PRL* 132.17 (2024)]. To make this point clearer in the main manuscript we have rephrased the end of the 4th paragraph:

We perform normal incidence X-ray PEEM, which is the optimum geometry for measuring both the in-plane Néel axis in the XMLD, and the altermagnetic XMCD. Images are taken at zero external field, where the XMCD signal due to the weak relativistic remanent \mathbf{M} is negligible compared to the altermagnetic XMCD due to $\mathbf{L} \parallel \langle 1-100 \rangle$ directions in the (0001)-plane [12]. The latter gives rise to our measured XMCD-PEEM contrast as confirmed by its spectral dependence (see Methods and Supplementary Information).

- Figure 2: this is not a vortex pair, but an antivortex pair. Looking at all patterned samples, it seems that with a global field and a closed surface, only anti-winding of 720 degrees can be achieved. Is there any way to obtain vortices, or are they limited to antiwinding / antivortex structures?

Answer

We thank the referee for pointing out that the structures shown in Fig.2 of the main manuscript form an antivortex pair and not a vortex pair. We have changed the text in the main manuscript accordingly.

The filled hexagon microstructures presented in the main manuscript are restricted by symmetry to form 720° anti-winding domain structures in the field-cooled state, as the referee has identified. In a **new section of the supplementary material, S9**, we present triangle microstructure geometries that form, by symmetry, isolated Bloch-type vortices (360° winding) in the field-cooled state, and whose vorticity is

controlled by the triangle orientation. This further highlights the precise control, through combined geometry restrictions and field-cooling, of the magnetic textures in our patterned structures.

- The authors should take care of including scale bars and correct colour bars in figures. For example, in Fig 3, it is difficult to tell the difference between the XMCD and XMLD images from the figure.

Answer

We thank the referee for bringing this to our attention. We have added spatial scale bars to all X-PEEM images in Figs.1-3. We have also made a clearer distinction between the XMLD- and XMCD-PEEM images of the unfilled hexagon device, shown in Fig.3a,b.

Referee #2 (Remarks to the Author):

I appreciate the efforts by the authors to revise their MS based on the comments by the reviewers. I read the entire MS again and I am still not convinced that the major issue, ie. overselling their interesting findings has been addressed appropriately.

I strongly recommend to try one more time to focus on the actual findings of this work, which are interesting on their own and lower the language to avoid the notion of overselling it.

I also feel that the broad brush of connection fundamental aspects with applications is in most cases more confusing than helpful.

For example: "For spintronics, altermagnets can merge favorable characteristics of conventional ferromagnets and antiferromagnets, considered for a century as mutually exclusive." This is a true statement, but there is more than spintronics, and the basic nature of this work does not need the connection to potential applications.

Answer

We highlight the importance of altermagnetism from the condensed matter physics perspective in the opening sentence. Because Nature is a broad audience journal, in our opinion it is important to also highlight the relevance of altermagnets from the perspective of microelectronics (spintronics). Since the referee confirms that our statement is true, and considering the broad audience of Nature, our preference is to keep it in the manuscript.

At this stage, I would recommend to focus on the following aspects

- the abstract is still confusing. I understand that the authors try to justify their work by relating to high level aspects, but what comes across is a mix of basic principles and phenomena with experimental efforts and technological applications in a way that is hard to follow unless the reader knows already what the authors want to say. It does not really summarize the actual findings of this work, but tries to create a story to "catch it all", which still gives the impression of "overselling".

Answer

In the revised abstract we removed all terms that may give an impression of overselling: In the third sentence we removed "severe" and "prominent", in the eighth sentence we removed "rich", and in the last sentence we removed "broad". We also revised the first sentence as follows: "Nanoscale detection and control of the magnetic order underpins a spectrum of condensed-matter research and device functionalities involving magnetism." We appreciate the referee's concern that the abstract does not give a detailed summary of the findings of the work, but in our opinion our revised abstract is consistent with Nature's guidance for authors: "Provide a general introduction to the topic and a brief non-technical summary of your main results and their implication".

- Given that the actual MS is rather short, I highly recommend to move parts from the SI section into the main text. There are two instances in the main text, which I suggest to consider

-- p4 spectral dependence of the data. This is important information and should be included in the Main part

Answer

We have moved the spectral dependence of the XMCD from the Supplementary Information to Figure 1(g) of the main manuscript. The following accompanying text to this panel has also been added to provide a direct reference to supplementary section S3 for the contrast reversal:

The Mn $L_{2,3}$ X-ray absorption and altermagnetic XMCD spectra are shown in Fig. 1g. The XMCD-PEEM images are obtained at fixed energy corresponding to the peak in the altermagnetic XMCD at the L_2 edge. The XMCD contrast reverses between positive and negative peaks of the XMCD spectrum, as shown in the Supplementary Information.

-- p6 stabilization of isolated Bloch-type vortices: this is a major result of this work and needs to be explained in the main text.

We thank the referee for this recommendation. We have moved the images of the isolated Bloch-type vortices from the Supplementary Information to panels j-l of Fig. 2 in the revised main manuscript, with the following explanation in the text:

In Figs. 2j-l, we show that the field-cooled state of a triangle microstructure can stabilize isolated Bloch-type vortices, whose chirality is controlled by the triangle orientation. The different topological textures arise due to the combination of the edge effect aligning the Néel vector with respect to the edge, and the external magnetic field selecting its sign. For the 60° separated edges of the hexagons, adjacent domains are rotated by 120° , while for the 120° separated edges of the triangles, adjacent domains are rotated by 60° . This highlights the important role of both the shape and scale of the patterned microstructures in the control of the magnetic configurations of antiferromagnetic MnTe.

Before I am able to make recommendation, I would like to see those concerns being addressed.

Referee #3 (Remarks to the Author):

Review of Amin et al., “Altermagnetism imaged and controlled down to the nanoscale”

In my opinion, the authors have generally answered the concerns of the referees, and have satisfied many of my concerns, both with added explanations, and additional data presented in the SI.

However, while the reply is very informative, the authors have not made significant changes to the manuscript itself – in particular, many of the physical insights that they now have shown in the referee reply / SI are missing from the manuscript. The figures have barely changed. They should properly revise the manuscript to improve the quality and include additional physical insights (including spectra, lengthscales, domain wall profiles, and the nucleation of vortices in triangles). Once they have done this, I would be willing to consider it for publication in Nature.

Specific comments:

- I appreciate that the authors have slightly modified their abstract, intro and conclusions to address the previous concerns. I feel it is much more representative of the work.

Answer

We thank the referee for their support.

- The discussion and extra data describing relevant lengthscales is appreciated. Normally such lengthscales are defined by a competition of competing factors, can they provide some discussion of this? How does it relate to the competition between the magnetoelastic and magnetocrystalline anisotropy?

Answer

We agree that the lengthscale over which the edge energy effect propagates will be a competition of energy terms that include the intrinsic magnetocrystalline anisotropy (MCA). A detailed study of the relative magnitude and function of the edge effect on the magnetic anisotropy is ongoing, but at this stage what is clear is that the edge effect is large enough to overcome the intrinsic magnetocrystalline anisotropy over approximately $\sim 1.7\mu\text{m}$.

We have added the following text to the main manuscript:

The edge effect is large enough to overcome the intrinsic magnetocrystalline anisotropy over a distance up to $\approx 1.7\mu\text{m}$ from the edge (see Supplementary Information), where the lengthscale is governed by the interplay of anisotropy, exchange and destressing energies [34].

- They now attribute the shape anisotropy to magnetoelastic interactions. However, no mapping of the strain in the sample is given. This should be possible to calculate or simulate. Can they do this, to provide additional insight / evidence as to the mechanism of the shape anisotropy?

Answer

As previously, we attribute the impact of the pattern edges on the magnetic domains as arising from a magnetoelastic interaction in a thin film strained on a substrate. More properly it could be described as an elastic energy term due to magnetostriction of the compensated magnet and the film-substrate clamping (discussed in detail in Ref. 34). We modified the description in the main text accordingly:

We utilize a known edge effect, arising from an elastic energy term due to magnetostriction of the film and film-substrate clamping, which can result in alignment of the \mathbf{L} -vector with respect to a patterned edge of a compensated magnet [31-34].

MnTe on InP is tensile strained (biaxial) due to the thermal expansion mismatch and the resulting strain which arises from the cooling from growth temperature [Kriegner et al., PRB 96, 214418 (2017)]. At the patterned edge, this strain will relax over a distance comparable to the film thickness [King et al., PRB 83, 115312, 7, 42107 (2011)]. However, the origin of this edge energy is not directly mapped onto any strain field but rather a more complex interaction between micromagnetic parameters and magnetoelastic charges as discussed in detail in Ref. 34. Furthermore, since the elastic parameters for MnTe thin films

grown on InP substrates are not well known, we do not see the value to adding a simulated strain map for our layers but note that we do not expect it to be qualitatively different from the many other systems in which this edge-induced anisotropy has been studied.

- I appreciate the authors including the discussion of the origin of the XMCD. They could still make this discussion a little more accessible to non-experts.

Answer

We have included additional clarification after the sentence:

Furthermore, the XMCD spectral shape due to L pointing in the (0001)-plane is qualitatively distinct from the XMCD spectral shape due to a net magnetization $M = M_1 + M_2$ along the [0001]-axis [12].

as follows:

This was demonstrated in Ref. [12] by comparing the measured XMCD spectral shapes at a zero magnetic field and at a 6T field applied along the [0001]-axis. In the former case, M is weak and the measured spectral shape agrees with the predicted spectral shape due to L. In the latter case, M is sizable and qualitatively modifies the spectral shape, again in agreement with theory.

- I appreciate the authors adding a spectrum to distinguish the XMCD arising from m and L. Is this spectrum measured in the PEEM on the same sample? The corresponding XAS spectrum should be plotted as well, to show the relative energy of the XMCD peaks with respect to the absorption edge.

Answer

To address this point, we have moved the XMCD spectrum from Supplementary Information to Figure 1g of the revised main text and included the corresponding XAS. The measured XAS and XMCD is from a different chip which is cut from the same wafer of MnTe material. The data was taken at a dedicated spectroscopy beamline (Diamond Light Source I06 branchline). The MAXPEEM beamline used for this study is not well suited to perform detailed energy dependent XAS/XMCD spectroscopy but rather utilises fixed energies with variable polarization for XMCD and XMLD. We have added the following clarification to the Methods section of the manuscript:

The XAS and XMCD spectra shown in Fig. 1g were obtained at beamline I06-1 of Diamond Light Source, from a different chip cut from the same wafer of MnTe material.

- The added analysis of the structure, showing the width of the domain walls is appreciated. Should be included in the main figures and text.

Answer

We have moved the analysis of the domain wall widths in the open hexagon microstructures from the Supplementary Information to Figure 4 of the main manuscript, and added the following text description:

In Fig.~4, we examine the domain wall profiles in the zero-field-cooled state of the unfilled hexagons. For the XMLD and XMCD measurements, the dependence of the signal on distance d across a 180° domain wall is described by functions $\text{sech}^2(2d/w)$ and $\tanh(d/w)$, respectively. The domain wall width parameter obtained for the fitted curves in Fig. 4b,d is $w=(134\pm 5)$ nm for the XMLD image, and (122 ± 13) nm for the XMCD image. Further analysis of domain wall profiles in unpatterned regions is included as Supplementary Information.

- The patterning / nucleation of vortices with triangle shapes, and antivortices with hexagonal shapes, is very nice. Should be added to the main text. Can the authors comment on why the symmetry of the hexagon or triangle leads to different topological textures? I.e. what is the mechanism for this?

Answer

We have moved the images showing nucleation of vortices in the triangle structures from Supplementary Information to the main manuscript in Figure 2j-l, and included the following discussion:

In Fig. 2j-l, we show that the field-cooled state of a triangle microstructure can stabilize isolated Bloch-type vortices, whose chirality is controlled by the triangle orientation. The different topological textures arise due to the combination of the edge effect aligning the Néel vector with respect to the edge, and the external magnetic field selecting its sign. For the 60° separated edges of the hexagons, adjacent domains are rotated by 120° , while for the 120° separated edges of the triangles, adjacent domains are rotated by 60° . This highlights the important role of both the shape and scale of the patterned microstructures in the control of the magnetic configurations of altermagnetic MnTe.

- Control: this seems to be a disagreement about wording: I am still not convinced. I see the micropatterning as a nice way to form particular structures/ states, e.g. vortices domain walls, and so they have control of the configuration via shape anisotropy, analogous to simple ferromagnets. But they do not have control of altermagnetic textures such as vortices and domain walls, as they cannot manipulate them.

Answer

We agree that this disagreement is largely about wording. However, we would like to point out that it is the combination of patterning with field cooling that gives us control of the formation of magnetic features, and part of this is reversible. While we can see that the patterning is not a reversible change, switching the field cool direction reverses the magnetic domains and textures. It allows us to choose the exact Néel vector domain orientation and the vortex and domain wall chirality. This is in some ways analogous to a ferromagnet but in others it exceeds it. For example, creating a stable vortex with a chosen chirality is completely new for a compensated magnet but also not simple in a ferromagnet which requires demagnetising procedures.

We also would like to reiterate that we have changed, in the previous version, five occurrences of “control” in the main manuscript to “controlled formation”, including in the abstract, concluding paragraph and Figure 2 caption title. This emphasizes that we are controlling the formation of the altermagnetic textures rather than, say, manipulating them electrically.

- “The X-ray dichroism vector mapping used here can be combined with other imaging techniques, such as X-ray laminography, potentially offering depth sensitivity and even higher spatial resolution [35].” – laminography is a similar technique to transmission tomography (simply a slightly different geometry). While it is true this could provide depth resolution, it does not provide higher spatial resolution. Perhaps the authors mean holography or ptychography, which offer diffraction limited spatial resolution?

Answer

We thank the referee for raising this point and have accordingly altered this sentence in the main manuscript to read “such as X-ray laminography or ptychography”.